# Crystal structure of inhibitor-bound human MSPL that can activate high pathogenic avian influenza

Ayako Ohno[1,*], Nobuo Maita[2,*] , Takanori Tabata[3], Hikaru Nagano[4] , Kyohei Arita[5], Mariko Ariyoshi[6] , Takayuki Uchida[1], Reiko Nakao[1], Anayt Ulla[1] , Kosuke Sugiura[1,7], Koji Kishimoto[8], Shigetada Teshima-Kondo[4], Yuushi Okumura[9] , Takeshi Nikawa[1]

**Infection of certain influenza viruses is triggered when its HA is cleaved by host cell proteases such as proprotein convertases and type II transmembrane serine proteases (TTSP). HA with a monobasic motif is cleaved by trypsin-like proteases, including TMPRSS2 and HAT, whereas the multibasic motif found in high pathogenicity avian influenza HA is cleaved by furin, PC5/6, or MSPL. MSPL belongs to the TMPRSS family and preferentially cleaves [R/K]-K-K-R↓ sequences. Here, we solved the crystal structure of the extracellular region of human MSPL in complex with an irreversible substrate-analog inhibitor. The structure revealed three domains clustered around the C-terminal α-helix of the SPD. The inhibitor structure and its putative model show that the P1-Arg inserts into the S1 pocket, whereas the P2-Lys and P4-Arg interacts with the Asp/Glu-rich 99-loop that is unique to MSPL. Based on the structure of MSPL, we also constructed a homology model of TMPRSS2, which is essential for the activation of the SARS-CoV-2 spike protein and infection. The model may provide the structural insight for the drug development for COVID-19.**

## Introduction

Mosaic serine protease large form (MSPL), and its splice variant TMPRSS13, was originally identified from a human lung cDNA library and is a member of the type II transmembrane serine proteases (TTSPs) (Kim et al, 2001; Kido & Okumura, 2008). TTSPs comprise a transmembrane domain near the N-terminus and a trypsin-like serine protease domain (SPD) at the C-terminus. Human

MSPL is expressed in lung, placenta, pancreas, and prostate (Kim et al, 2001). MSPL is reported to cleave the spike protein of porcine epidemic diarrhea virus (Shi et al, 2017), MERS- and SARS-CoV (Zmora et al, 2014), certain influenza virus HAs (Okumura et al, 2010), and pro-hepatocyte growth factor (Hashimoto et al, 2010), but the physiological function of MSPL is poorly understood. TTSPs share a similar overall organization comprising an N-terminal cytoplasmic domain, transmembrane region, and stem/catalytic domains at the C terminus (Szabo & Bugge, 2008). All TTSPs are synthesized as single-chain zymogens and are subsequently activated into the two-chain active forms by cleavage within the highly conserved activation motif. The two chains are linked by a disulfide bridge so that TTSPs remain bound to the cell membrane (Bugge et al, 2009). The catalytic domain contains a highly conserved "catalytic triad" of three amino acids (His, Asp, and Ser). Like all other trypsin-like serine proteases, MSPL possesses a conserved Asp residue on the bottom of the S1 substrate-binding pocket; therefore, it accepts substrates with Arg or Lys in the P1 position. Based on similarities in the domain structure, the SPD, TTSPs are classified into four subfamilies: hepsin/TMPRSS, matriptase, HAT/DESC, and corin (Szabo & Bugge, 2008, 2011; Antalis et al, 2011; Böttcher-Friebertshäuser, 2018). MSPL belongs to the hepsin/TMPRSS subfamily. In this subfamily, hepsin and spinesin (TMPRSS5) contain a single scavenger receptor cysteine-rich repeat (SRCR) domain in the stem region, whereas MSPL, TMPRSS2, -3, and -4 contain an additional low-density lipoprotein receptor A (LDLA) domain near the single SRCR domain in the stem region (Antalis et al, 2011; Szabo & Bugge, 2011). Furthermore, enteropeptidase has additional insertions of SEA, LDLA, CUB, MAM, and CUB domains between the transmembrane and the LDLA domain (Kitamoto et al, 1994). The SRCR domain has ~100–110 amino acids that adopt a compact fold consisting of a curved

[1]Department of Nutritional Physiology, Institute of Medical Nutrition, Tokushima University Graduate School, Tokushima, Japan [2]Division of Disease Proteomics, Institute of Advanced Medical Sciences, Tokushima University, Tokushima, Japan [3]Laboratory for Pharmacology, Pharmaceutical Research Center, Asahikasei Pharma, Shizuoka, Japan [4]Department of Nutrition, Graduate School of Comprehensive Rehabilitation, Osaka Prefecture University, Osaka, Japan [5]Graduate School of Medical Life Science, Yokohama City University, Kanagawa, Japan [6]Graduate School of Frontier Biosciences, Osaka University, Osaka, Japan [7]Department of Orthopedics, Institute of Biomedical Sciences, Tokushima University Graduate School, Tokushima, Japan [8]Graduate School of Technology, Industrial and Social Sciences, Tokushima University, Tokushima, Japan [9]Department of Nutrition and Health, Faculty of Nutritional Science, Sagami Women's University, Kanagawa, Japan

Correspondence: okumura_yushi@isc.sagami-wu.ac.jp
Ayako Ohno's present address is Curreio Inc., Tokyo, Japan Nobuo Maita's present address is Institute for Quantum Life Science, National Institute for Quantum and Radiological Science and Technology, Chiba, Japan
*Ayako Ohno and Nobuo Maita contributed equally to this work

β-sheet wrapped around an α-helix, and is stabilized by 2–4 disulfide bonds. Depending on the number and the position of the cysteine residues, the SRCR domain has been divided into three subclasses (group A, B, and C) (Ojala et al, 2007). The canonical LDLA domain contains ~40 amino acids and contains six conserved cysteine residues that are involved in the formation of disulfide bonds. The LDLA domain also contains a calcium ion coordinated with six highly conserved residues near the C-terminus. Together, the disulfide bonds and calcium-binding stabilize the overall structure of the LDLA domain (Daly et al, 1995).

Limited proteolysis of the glycoprotein on the viral surface mediated by a host protease is a key step in facilitating viral infection. The influenza viral HA0 is cleaved by various host proteases and divided into HA1 and HA2 subunits, where HA1 mediates host cell binding as well as the initiation of endocytosis and HA2 controls viral-endosomal membrane fusion (Hamilton et al, 2012). Previous studies show that TMPRSS2, -4, DESC1, HAT, and MSPL activate the influenza virus by cleaving HA0 (reviewed in Böttcher et al [2006]; Chaipan et al [2009]; Okumura et al [2010]; Antalis et al [2011]; Ohler and Becker-Paul, [2012]; Böttcher-Friebertshäuser et al [2013]; Zmora et al [2014]; Böttcher-Friebertshäuser [2018]). A newly synthesized HA is cleaved during its transport to the plasma membrane in the trans-Golgi network by furin or TMPRSS2, whereas HAT cleaves it at the cell surface during viral budding (reviewed in Böttcher-Friebertshäuser [2018]). There are two types of cleavage site sequences; monobasic motifs have single or discrete basic residues such as [Q/E]-[T/X]-R↓ or R-X-X-R↓ (vertical arrow indicates the cleavage position), and multibasic motifs are composed of Lys/Arg-rich sequences such as R-X-[K/R]-R↓. The multibasic motif is found in highly pathogenic avian influenza virus, and is mainly cleaved by furin, PC5/6 (Stieneke-Gröber et al, 1992; Horimoto et al, 1994), and MSPL (Kido et al, 2009). Some highly pathogenic avian influenza virus variants, such as H5N2 (Lee et al, 2005) and H7N3 (Bulach et al, 2010) have the multibasic motif with Lys at the P4 position (K-K-K-R↓), and these HA proteins are not cleaved by furin nor PC5/6, but MSPL (Thomas, 2002; Remacle et al, 2008; Kido et al, 2009). Therefore, MSPL is a key protease to protect humans from an outbreak of novel avian influenza A virus.

To date, the extracellular region of human hepsin (Somoza et al, 2003; Herter et al, 2005) and serine protease domain (SPD) of enteropeptidase (Lu et al, 1999; Simeonov et al, 2012) are the only structures reported among the hepsin/TMPRSS family. The crystal structure of hepsin revealed that the SRCR domain is located at the opposite side of the active site of SPD, and these domains are splayed apart. Because hepsin lacks the LDLA domain, the relative orientation of the LDLA, SRCR, and SP domains in other members of the hepsin/TMPRSS family is unknown. To elucidate the spatial arrangement of the three domains and multibasic motif recognition, we determined the crystal structure of the extracellular region of human MSPL in complex with the irreversible peptidic inhibitor decanoyl–RVKR–cmk at 2.6 Å resolution.

To our surprise, the overall structure of MSPL reveals that the spatial arrangement of the SRCR and SP domains in MSPL is markedly different from that of hepsin. Although the inhibitor adopts an artificial conformation because of crystal packing, the predicted peptide model explains how MSPL is able to recognize both Lys or Arg as P4 residues. In addition, we constructed a homology model of human TMPRSS2, which is reported to cleave the spike protein of SARS-CoV-2 (Bestle et al, 2020; Hoffmann et al, 2020a, 2020b). The human TMPRSS2 model reveals a wide binding cleft at the S1' position, suggesting that TMPRSS2 can capture the target peptides of flexible conformations.

# Results

## Overall structure of the human MSPL extracellular region

The extracellular region of human MSPL is composed of an LDLA domain (residues 203–226), an SRCR domain (residues 227–317) and a SPD (residues 326–561) (Fig 1A). We expressed and purified the extracellular region (residues 187–586) of human MSPL and crystallized the protein with decanoyl–RVKR–cmk, which is known as furin inhibitor I. Diffraction data were collected at the Photon Factory AR-NE3a and the structure was solved to a resolution of 2.6 Å (Fig 1B and Table S1). This is the first published structure of an LDLA-containing hepsin/TMPRSS subfamily protein. The refined model contains the human MSPL with the residue range of 193–563, decanoyl–RVKR–cmk, and a calcium ion (Fig S1B). Residues of 187–192, 324–325, and 564–586 regions were missing because of the disorder. There are four potential $N$-glycosylation sites, and two $N$-glycans attached to Asn255 and Asn405 were observed, but no phosphorylated residues were found (Murray et al, 2017).

The extracellular region of human MSPL is composed of the non-catalytic portion of the N-terminal region (LDLA and SRCR domain) and the SPD at the C-terminus (Fig 1B). The three domains are linked to each other by disulfide bonds. The human MSPL is activated by hydrolytic cleavage at Arg325-Ile326 then residues in the 326–586 region are converted to the mature SPD (Okumura et al, 2010). Ile326 (Ile16 in chymotrypsin; hereafter, the residue numbers in parentheses denote the corresponding chymotrypsin residue number, see Fig 3B) is located in a pocket where the N atom interacts with the side chain of Asp510(194) (Fig S1A). Therefore, this structure could represent the active form in which human MSPL is processed by an intrinsic protease during expression in the cell. The LDLA domain of human MSPL is 24 residues in length and composed of two turns and a short α-helical region. A canonical LDLA domain has an N-terminal antiparallel β-sheet and three disulfide bonds (Daly et al, 1995). Therefore, the LDLA of human MSPL lacks half of the canonical N-terminal region. Because the SRCR domain of human MSPL has only two disulfide bonds, it does not belong to either group A or B (Sarrias et al, 2004). Intriguingly, the 3D structures of the SRCR domains of human MSPL and hepsin are very similar despite their low level of sequence homology (23% sequence identity), suggesting that the SRCR domain of MSPL belongs to group C (Ojala et al, 2007).

To date, 3D structures of SRCR-SPD of hepsin (Protein Data Bank [PDB] entry: 1P57 & 1Z8G) and SPD of enteropeptidase (PDB entry: 1EKB & 4DGJ) have been reported in the same hepsin/TMPRSS subfamily. Here, we compared the structures of human MSPL and hepsin (Figs 1A and 2A–C). The root-mean-square deviation of the two SPDs (r.m.s.d. of Cα atoms = 0.637 Å), as well as the SRCR domains (r.m.s.d. of Cα atoms = 0.988 Å), are quite small. Although

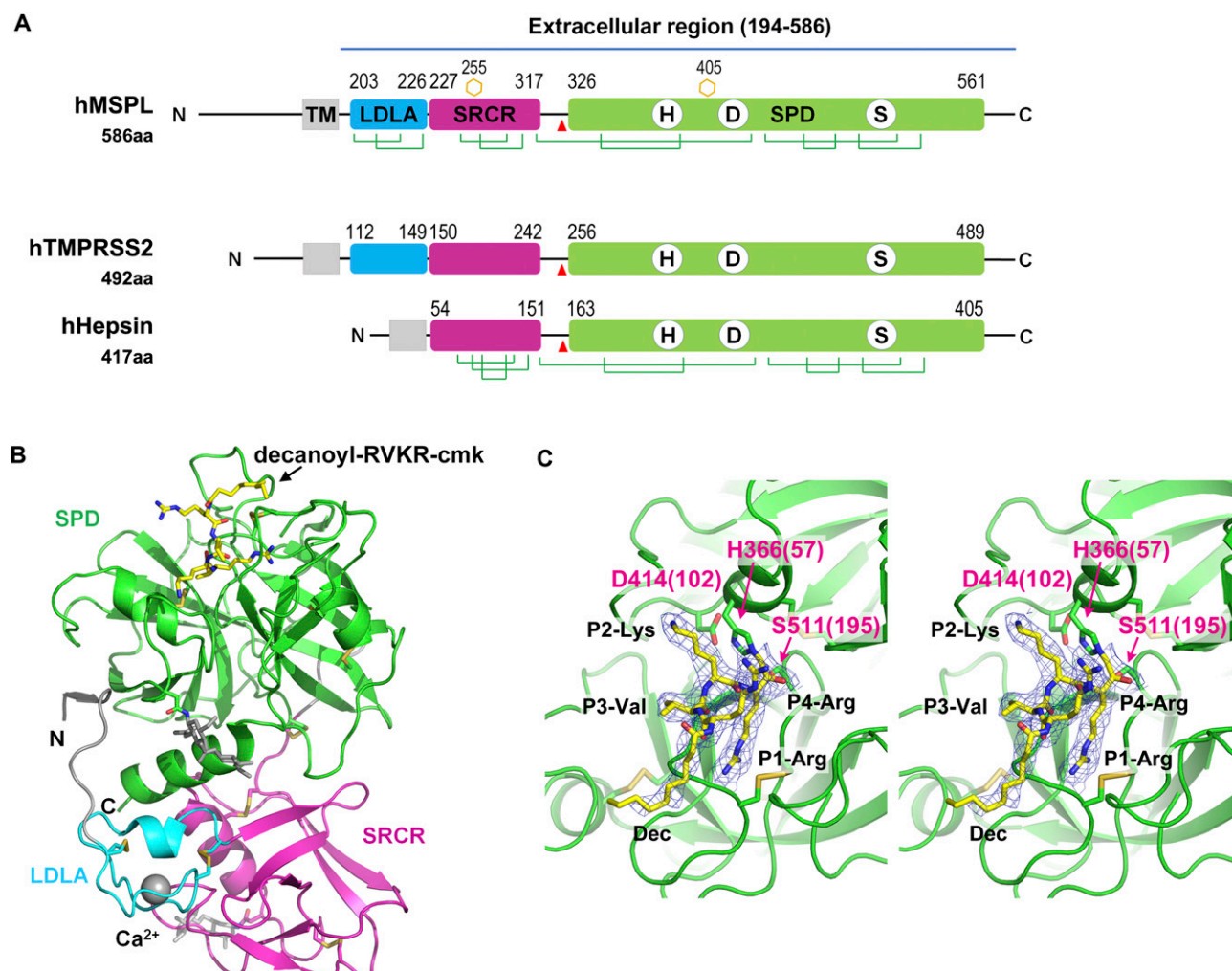

Figure 1.  Overall structure of the human MSPL extracellular domain.
**(A)** Schematic illustration of full-length human MSPL. Human MSPL is composed of a cytoplasmic region (1–165), transmembrane helix (166–186), truncated LDL-receptor class A (LDLA) domain (203–226), scavenger receptor cysteine-rich (SRCR) domain (227–317), and serine-protease domain (SPD) (326–561). Human MSPL is autocleaved at after the Arg325 (red arrowhead) to generate the mature protein form. *N*-glycosylated Asn observed in the crystal structure are shown as orange hexagons. Disulfide bonds are shown as green lines. To compare the representative proteins of hepsin/TMPRSS family, human TMPRSS2 and hepsin are also shown. **(B)** Ribbon representation of the crystal structure of the human MSPL extracellular region covalently bonded with decanoyl–RVKR–cmk (yellow stick model). LDLA domain (cyan), SRCR domain (magenta), and SPD (green) are shown. LDLA domain binds Ca²⁺ in the center of the loop. The N-terminal region (194–196) interacts with SPD by making a β-sheet. Two *N*-glycans were observed at Asn255 and Asn405 (white stick model). **(C)** A close-up view of bound decanoyl–RVKR–cmk inhibitor and the catalytic triad with the wall-eyed stereo presentation. Numbers in parentheses indicate the corresponding residue number of chymotrypsin. The refined 2Fo-Fc electron density map (contoured at > 1σ) of the inhibitor is shown.

the SPD and SRCR domains of human MSPL and hepsin are almost identical, the arrangement of each domain with respect to each other is markedly different (Fig 2B). Specifically, when the 3D structures of SPD from hepsin and human MSPL are fitted, the SRCR domain of MSPL is rotated by ~80 degrees relative to that of hepsin. The difference may be caused by the presence of the LDLA domain in human MSPL. The LDLA, SRCR, and SP domains of human MSPL are more tightly packed than in hepsin, where these domains are splayed apart. Accordingly, a short parallel β-sheet between the N-terminal segment and the SP domain was observed in human MSPL, whereas the C-terminal end of hepsin forms an antiparallel β-sheet (Fig 2A).

There are only six residues between the transmembrane domain and the N-terminal Thr193 residue of our structural model. Hence, the extracellular region of human MSPL must be located very close to

the plasma membrane. Indeed, the region that was predicted to be close to the plasma membrane is enriched in basic residues, such as Arg196, Lys198, Lys218, Lys220, and Arg561(245) (Fig 2C). The extracellular region of hepsin is also thought to lie flat against the plasma membrane (Somoza et al, 2003). Hence, both MSPL and hepsin may bind substrate in close proximity to the transmembrane region. However, the extracellular region of human MSPL is oriented in the opposite way with respect to that of hepsin Supplemental Data 1.

### Interaction of the inhibitor decanoyl–RVKR–cmk in the active site of human MSPL

As expected, the SPD of human MSPL displays the conserved architecture of the trypsin- and chymotrypsin-like (S1A family) serine

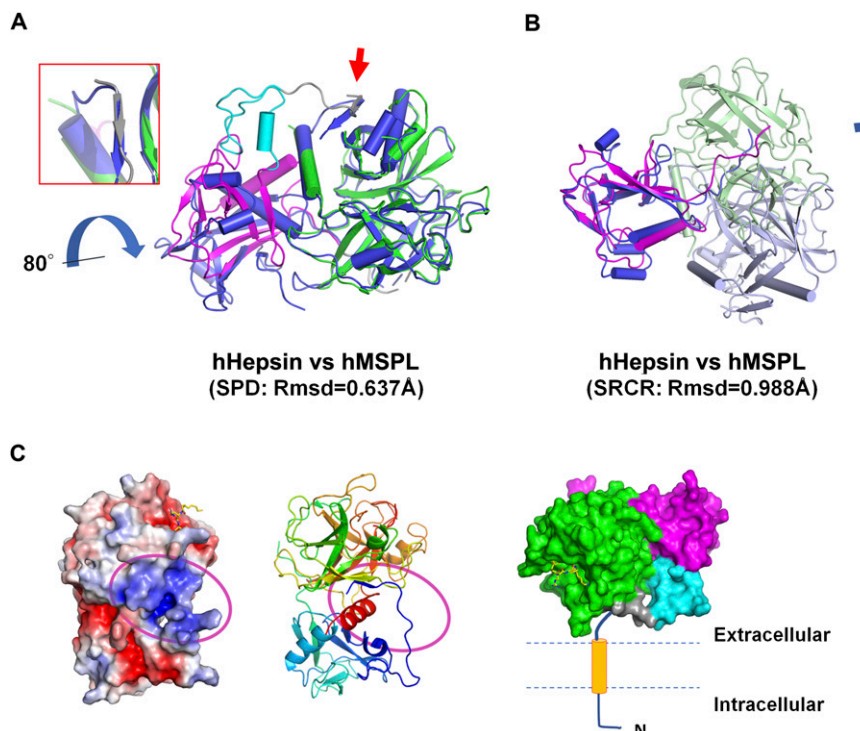

**Figure 2. Comparison of human MSPL and hepsin.**
**(A)** Human Hepsin (coloured in marine blue) and human MSPL (coloured in cyan [LDLA], magenta [SRCR], and green [SPD]) were superposed with the SPD. The RMSD value is 0.637 Å calculated with 197 Cα atom positions. A β-sheet interaction of the N-terminus and SPD in MSPL is replaced by the C-terminus in hepsin (red arrow and close-up view in the red box). The hepsin SRCR domain is rotated by about 80° relative to that of human MSPL. **(B)** Human Hepsin (coloured in blue [SRCR] and pale blue [SPD]) and human MSPL (coloured in magenta [SRCR] and pale green [SPD]) were superposed with the SRCR domain. The RMSD value is 0.988 Å calculated with 59 Cα atom positions. **(C)** (Left) The electrostatic surface potential of the human MSPL extracellular domain. A characteristic positively-charged patch (magenta oval), composed of Arg196, Lys198, Lys218, Lys220, and Arg561, is thought to act as a contact surface for the cell membrane. The potential map is coloured from red (−5kT/e) to blue (+5kT/e). Middle: A ribbon model of human MSPL is shown with the same orientation. Right: A putative model of the membrane-anchored full-length human MSPL coloured with green (SPD), cyan (LDLA), magenta (SRCR), and orange (transmembrane domain).

proteases (Fig 1B). In the activated human MSPL, Ile326(16) at the cleavage site forms a salt bridge with the conserved Asp510(194) residue located immediately before the catalytic Ser511(195) residue (Fig S1A). This interaction is generated by the activating cleavage (Stubbs et al, 1998) as observed in hepsin (Somoza et al, 2003) and other TTSP family members (Lu et al, 1999; Kyrieleis et al, 2007). Formation of the S1 pocket and oxyanion hole comes about via a conformational change in the nearby hairpin loop (189-loop) (Khan & James, 1998). This "Ile16-Asp194 salt-bridge" is a common feature among the trypsin-like proteases (Halfon & Craik, 1998). The chloromethyl group of the inhibitor irreversibly alkylates His366(57) of the SPD of human MSPL, in addition, a hemiketal if formed to the active site Ser511(195). In addition, several interactions via the P1-Arg and P2-Lys side chains are formed (Figs 1C and 3A). Covalent interaction between the decanoyl–RVKR–cmk inhibitor and catalytic residues (His366(57), Ser511(195)) occurs via a nucleophilic attack on the cmk moiety. P1-Arg inserts into the deep S1 pocket, and its carbonyl oxygen atom directly binds to the backbone amides of the oxyanion hole (Gly509(193) and Ser511(195)). In addition, the guanidino group of P1-Arg forms a salt bridge with the side chain of Asp505(189), as well as a hydrogen bond with the side chain of Ser506(190). Asp505(189) is located at the bottom of the S1 pocket. These residues are highly conserved among the hepsin/TMPRSS subfamily (Fig 7). The interaction between P1-Arg and human MSPL is characteristic of trypsin-like serine proteases. However, P2-Lys interacts with basic residues located at the 99-loop (chymotrypsin numbering) next to the catalytic residue Asp414(102). The Nζ of P2-Lys forms five hydrogen bonds with the backbones of Asp408(96) and Glu410(98), the side chains of Tyr406(94) and Asp411(99) and a water molecule. This water molecule also mediates hydrogen bonding interactions with the side chains of Asp411(99) and the catalytic Asp414(102) residue. Interestingly, with the exception of catalytic Asp414(102), residues that interact with the side chain of P2-Lys are not

conserved among the hepsin/TMPRSS subfamily (Fig 7, cyan dot). Compared with P1-Arg and P2-Lys, there are no distinct interactions between the side chains of P3-Val/P4-Arg and the human MSPL. The backbone carbonyl of P3-Val forms a hydrogen bond with the backbone amide of Gly532(216). The side chain of P3-Val makes hydrophobic interactions with Trp531(225) and Gly532(216). By contrast, the backbone of P4-Arg forms no hydrogen bonds with the human MSPL but with the Asp472(160) of crystallographic symmetrical subunit (see below). The N-terminal decanoyl moiety makes hydrophobic interactions with Gln537(221) at the 220-loop (chymotrypsin numbering). One ordered sulfate ion is located in close proximity to both P3-Val and P4-Arg where it forms hydrogen bonds with the backbone amides of P2-Lys and P3-Val.

Although the model is well fitted to the electron density (Fig 1C), the P3-P4 moiety of decanoyl–RVKR–cmk bound at human MSPL seems to be in an abnormal conformation compared with other substrate peptides bound to S1A family members (Perona & Craik, 1997; Herter et al, 2005; Debela et al, 2007). In most cases, the backbone nitrogen and oxygen atoms of the P3 residue interact with glycine (Gly216 in chymotrypsin) to form an antiparallel β-sheet interaction (Perona & Craik, 1997). However, the nitrogen atom at P3-Val does not interact with the oxygen atom at Gly532(216) (Fig 3A). Closer inspection of the structure reveals an abnormal conformation of the P3-P4 moiety, most likely arising from crystal packing. We observed that the guanidino group of P4-Arg tightly interacts with Asp472(160) in the symmetrical subunit, and the sulfate ion stabilizes the conformation (Fig 4A). Therefore, we suspect that the P3-P4 portion of the inhibitor peptide in our structure does not reflect the proper binding conformation. We therefore built a putative model of the target peptide based on the acetyl–KQLR–cmk structure bound to human hepsin (PDB entry: 1Z8G, Herter et al, 2005) (Figs 4B and C and S2A–D). In this model, the

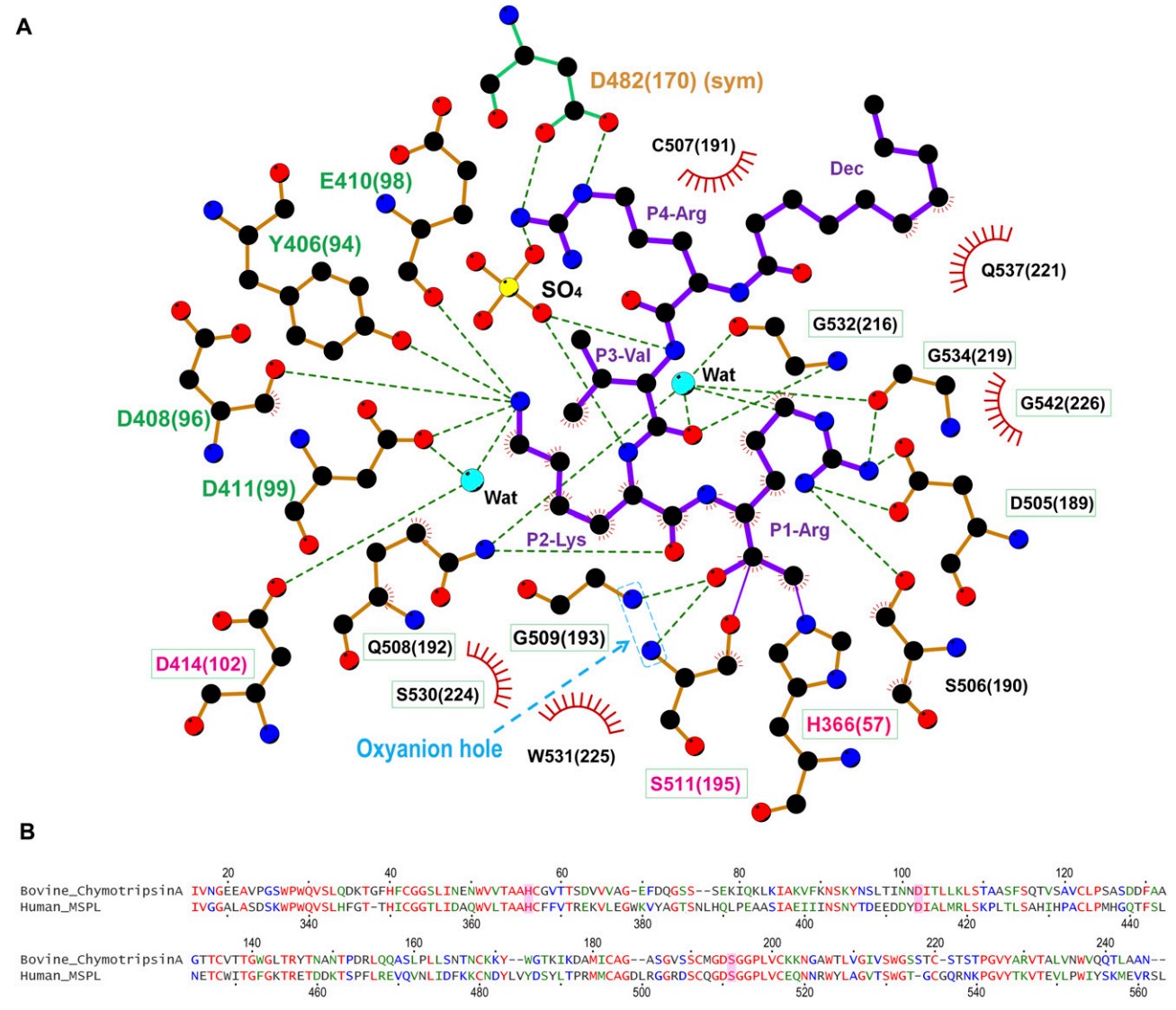

**Figure 3. Interaction of decanoyl–RVKR–cmk inhibitor with human MSPL.**
**(A)** The SPD of human MSPL and decanoyl–RVKR–cmk are shown in orange and purple, respectively. Nitrogen atoms, blue; oxygen atoms, red; carbon atoms, black; sulfur atoms, yellow. Dashed lines represent hydrogen bonds. Red semi-circles with radiating spokes denote the residues of the human MSPL involved in hydrophobic contacts with decanoyl–RVKR–cmk. Cyan spheres denote water molecules. Light-blue dashed rectangle denotes the oxyanion hole. The catalytic triad of three amino acids is highlighted in magenta. The residues that interact with P2-Lys and P4-Arg are highlighted in green and orange, respectively. Conserved residues among human MSPL, TMPRSS2-4, and hepsin are highlighted in green boxes. The figure was prepared with LigPlot+ (Laskowski & Swindells, 2011). **(B)** Sequence alignment of bovine α-chymotrypsin and SPD of human MSPL. The catalytic triad is highlighted as magenta.

guanidino group of P4-Arg is in close proximity to the negatively charged region around the 99-loop (Glu409(97), Glu410(98), Asp411(99), and Tyr489(175)). Because these residues are unique to MSPL, the structure may explain why this enzyme shows a target preference for the P4-Arg/Lys sequence.

### Comparison of the binding mechanisms of decanoyl–RVKR–cmk inhibitor to human MSPL and furin

The crystal structure of the decanoyl–RVKR–cmk inhibitor in complex with mouse furin has been determined (Henrich et al, 2003). Although furin also has the same Ser-His-Asp catalytic

triad as MSPL, its catalytic domain belongs to the superfamily of subtilisin-like serine proteases (Siezen & Leunissen, 1997). The catalytic domain of furin has a different overall fold from that of human MSPL, which belongs to the trypsin- and chymotrypsin-like (S1A family) serine protease family. Despite the different overall fold of human MSPL and furin, decanoyl–RVKR–cmk inhibits both enzymes. Therefore, we compared the structure of the human MSPL-bound decanoyl–RVKR–cmk inhibitor with that of the furin-bound inhibitor (Fig 5). Except for the P1-Arg and P2-Lys, they are not superimposed. In the human MSPL–decanoyl–RVKR–cmk complex structure, the inhibitor exhibits a bend at the P3-Val. By contrast, in the furin–decanoyl–RVKR–cmk complex structure, the inhibitor fits an extended

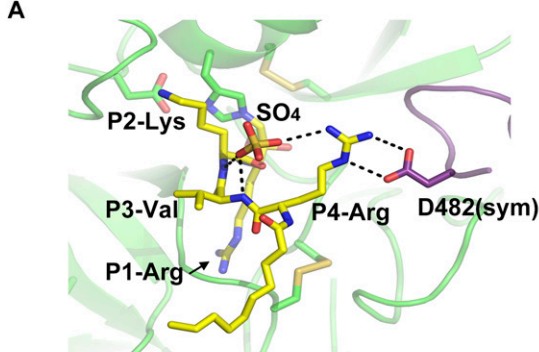

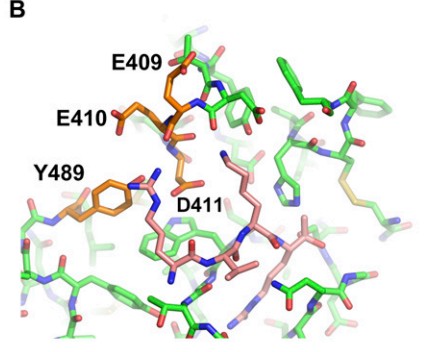
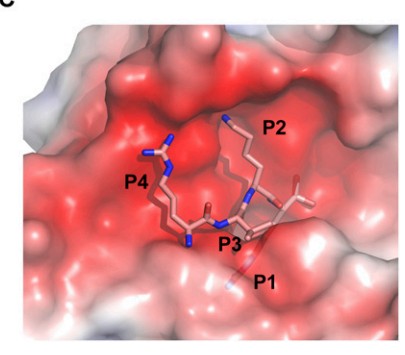

**Figure 4. Putative model of RVKR peptide bound to human MSPL.**
**(A)** Side chain of P4-Arg interacts with a sulfate and Asp482 in a symmetry-related subunit (purple). **(B)** Putative RVKR peptide was modelled with acetyl–KQLR–cmk structure bound to human hepsin (PDB entry: 1Z8G) as template. P4-Arg interacts with acidic residues in the 99-loop (Glu409, Glu410, and Asp411) and with Tyr489. **(C)** Electrostatic surface potential of the MSPL SPD with the putative RVKR peptide (stick model in rose red). The potential map is coloured from red (−5kT/e) to blue (+5kT/e).

conformation. As a consequence, the P1, P2, and P4 site contacts with furin, whereas the P3 side chain is directed into the solvent. As described earlier, the P3 and P4 site of decanoyl–RVKR–cmk bound to MSPL is most likely an artifact. Compared with the putative model of the MSPL-bound inhibitor, the P1-P3 site is almost identical, whereas the side chain of P4-Arg is in the opposite orientation (Fig 5). As the S4 site of furin is optimized for Arg binding, furin shows reduced affinity for the P4-Lys sequence (Henrich et al, 2003). However, the S4 site of MSPL comprises a wide negatively charged surface that is suitable for multibasic motif binding (Fig 4C).

## Discussion

By the end of January 2021, the SARS-CoV-2 pandemic has killed more than 2.2 million people (https://ourworldindata.org/covid-deaths) and resulted in a worldwide recession as people were forced to socially distance. The infection of SARS-CoV-2 requires cleavage at the S1/S2 site by furin, followed by cleavage at the S2′ site by TMPRSS2 (Bestle et al, 2020; Hoffmann et al, 2020a, 2020b). TMPRSS2 displays low sequence preference for the P2-P4 position (Böttcher et al, 2006; Baron et al, 2013). The reason for TMPRSS2 mediated specific cleavage of the S2′ site is therefore unclear.

To date, the experimental structure of TMPRSS2 has not been reported. However, human MSPL shares 45% amino acid identity with TMPRSS2. Consequently, there is sufficient similarity to build a reliable homology model of human TMPRSS2 using MSPL as template (Fig 6 and Supplemental Data 1). Eight out of nine disulfide bonds are conserved (Fig 7), and the relative domain alignment of human TMPRSS2 is similar to that of MSPL. However, the SP domain,

specifically at the β12-β13 loop region (60-loop in chymotrypsin), displays significant differences (Fig 6). These structural changes result in a wide substrate-binding cleft (Fig 6), so that human TMPRSS2 may be able to capture the target peptides of not only a stretched conformation but also flexible conformations.

Furthermore, Asp411, one of the important residues for P2-Lys and P4-Arg recognition found in MSPL, is replaced by Lys225 in TMPRSS2 (Figs 4B and 5). This substitution leads to a reduced negatively charged region on the S3–S5 site (Fig 6). Nonetheless, the electrostatic surface potential of the S2 site in TMPRSS2 is still negatively charged (Fig 6) and able to participate in P2-Lys binding.

Our structure also helps to predict the tertiary structure of TMPRSS3, the gene responsible for autosomal recessive non-syndromic deafness. Mutations identified in patients with this syndrome were mapped onto a homology model of TMPRSS3 to better understand the disease. Seven missense TMPRSS3 mutants (D103G, R109W, C194F, R216L, W251C, P404L, and C407R) associated with deafness in humans were unable to activate the ENaC (Wattenhofer et al, 2005; Antalis et al, 2010). One of seven missense mutants associated with the loss of hearing, D103G, was found in the LDLA domain of human TMPRSS3 (Guipponi et al, 2002; Wattenhofer et al, 2005). Because Asp103 in human TMPRSS3 corresponds to Asp222 in human MSPL, the LDLA structure stabilized by calcium-binding may be important for the function of the protein (Fig S1B). Indeed, the mutations in LDLA and SRCR (D103G, R109W, and C194F) as well as the SPD of human TMPRSS3 affect its autoactivation by proteolytic cleavage at the junction site between the SRCR and the SP domains (Guipponi et al, 2002).

In summary, we have elucidated the structure of the extracellular domain of human MSPL and its spatial arrangement of three (LDLA,

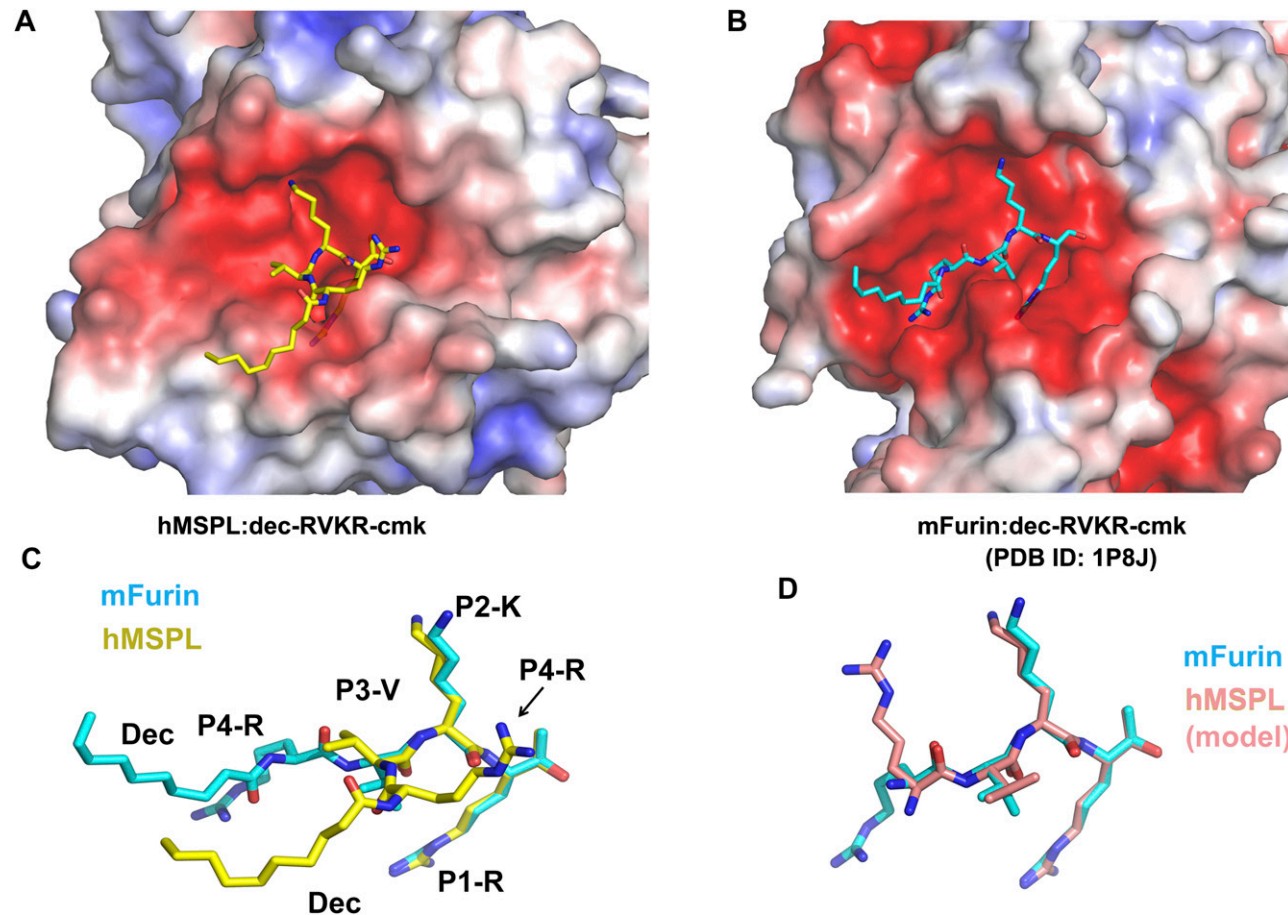

**Figure 5. Conformational differences between the decanoyl–RVKR–cmk inhibitor bound to human MSPL and mouse furin.**
**(A)** The human MSPL–decanoyl–RVKR–cmk inhibitor complex. Human MSPL and inhibitor are shown as an electrostatic surface potential representation and yellow stick model, respectively. The potential maps are coloured from red (−5kT/e) to blue (+5kT/e). **(B)** The mouse furin–decanoyl–RVKR–cmk inhibitor complex (PDB entry: 1P8J). Mouse furin and inhibitor are shown as an electrostatic surface potential representation and cyan stick model, respectively. The potential maps are coloured from red (−5kT/e) to blue (+5kT/e). **(C)** Superposition of decanoyl–RVKR–cmk inhibitors bound to human MSPL and mouse furin. **(D)** Superposition of putative RVKR inhibitors bound to human MSPL and mouse furin.

SRCR, and SP) domains, as well as the substrate sequence specificity of human MSPL. These findings will be useful in designing novel anti-influenza drugs that prevent HPAI virus uptake into the host cell. Human MSPL also contributes to the cleavage and activation of severe acute respiratory syndrome coronavirus (SARS-CoV) Middle East respiratory syndrome coronavirus (MERS-CoV) spike proteins (Zmora et al, 2014).

The mechanism of infection of SARS-CoV-2 needs to be elucidated as a matter of urgency. The MSPL structure shares the highest sequence homology to TMPRSS2 among the experimentally solved structures. The homology model presented in this article provides novel insight into TMPRSS2 function. However, it is still necessary to solve the structure of TMPRSS2.

## Note added in proof

Recent studies have shown that TMPRSS13/MSPL is involved in the cleavage of the spike protein of SARS-CoV-2 as the same extent as TMPRSS2 (Hoffman et al, 2021; Kishimoto et al, 2021).

# Materials and Methods

### Expression and purification of human MSPL

Soluble recombinant human MSPL was generated using a previously established stable cell line expressing human MSPL (Okumura et al, 2010), which accumulated in serum-free culture medium (SFCM). It should be noted that the human MSPL we used here is a splice variant (GenBank id: BAB39741) of the canonical sequence that includes the single amino acid substitution L586V. Approximately 10 L of SFCM was concentrated by ultrafiltration using a Pellicon XL 50 (Merck-Millipore). The resulting SFCM was applied to an Anti-FLAG M2 agarose gel equilibrated in 50 mM Tris–HCl and 150 mM NaCl, pH 7.4 (TBS). Bound protein was subsequently eluted in 0.1 M glycine–HCl, pH 3.5, and fractions containing recombinant human MSPL pooled and dialyzed into PBS.

### Complex formation, crystallization, and data collection

The peptide inhibitor (decanoyl–RVKR–cmk) was purchased from Merck-Millipore and reconstituted in DMSO. Human MSPL–inhibitor

A

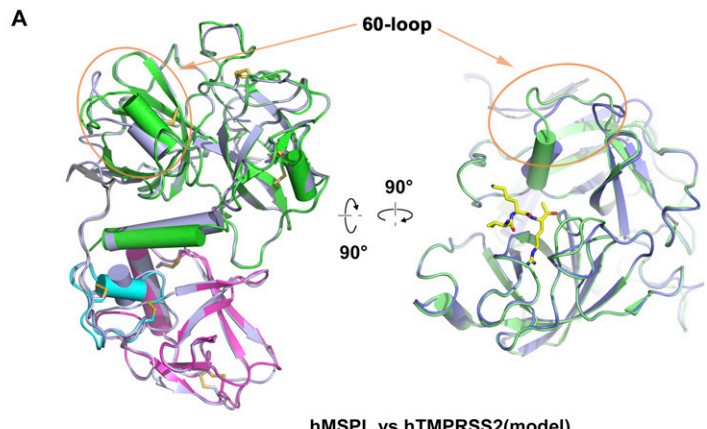

hMSPL vs hTMPRSS2(model)

B

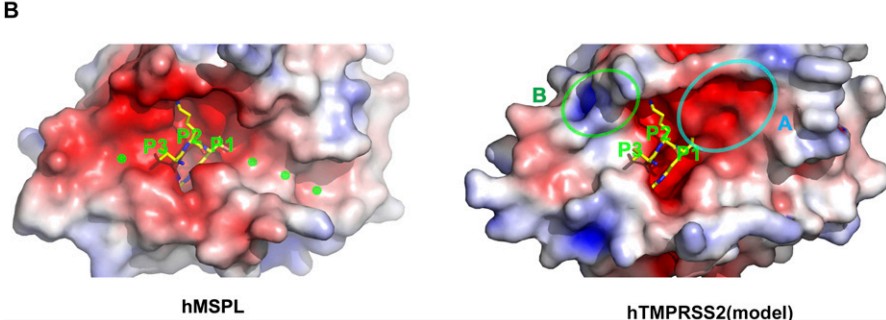

hMSPL                hTMPRSS2(model)

**Figure 6.  Homology model analysis of human TMPRSS2.**

**(A)** A homology model of human TMPRSS2 (gray ribbon) was built with human MSPL (LDLA [cyan], SRCR [magenta], and SPD [green]) structure as a template. Superposed analysis revealed large structural differences at the β12-β13 loop (60-loop) region (orange oval). The coordinate of the homology model of human TMPRSS2 is available from supplementary materials. **(B)** Electrostatic surface potential of human MSPL and human TMPRSS2 SPD. Left panel: Human MSPL has a narrow groove that fits with the stretched peptide chain (green dots). Right panel: Human TMPRSS2 has a wider cleft at the P1′ binding site (highlighted by the cyan oval A). A positively charged area derived from Lys225 is indicated by the green oval B. The potential map is coloured from red (–5kT/e) to blue (+5kT/e).

complex was formed by incubating purified human MSPL (6.1 mg/ml) with a fourfold molar excess of decanoyl–RVKR–cmk at 4°C for 5 min and then centrifuged (25,000*g*) at 4°C, for 5 min to remove the precipitate. Crystallization screening was performed by mixing 1 μl of the human MSPL–inhibitor solution with 1 μl of reservoir solution using the hanging-drop vapor diffusion method. The human MSPL–inhibitor complex was crystallized at 15°C with a reservoir solution composed of 0.1 M Hepes (pH 7.5), 2.4 M ammonium sulfate. Before data collection, the single crystal was transferred to the cryoprotectants (20% [vol/vol] glycerol and 80% [vol/vol] of the reservoir) for 5 s, and then flash-frozen in liquid nitrogen. The diffraction dataset for the human MSPL–decanoyl–RVKR–cmk crystal was collected at beamline NE3A in the Photon Factory Advanced Ring. The crystal belonged to space group $P2_12_12_1$ with unit cell parameters $a$ = 55.84, $b$ = 62.40, and $c$ = 171.63 Å. Diffraction data were processed using the program *iMosflm* (Battye et al, 2011), followed by *Aimless* (Evans & Murshudov, 2013). Data collection statistics are summarized in Table S1.

**Structure determination and refinement of the human MSPL–inhibitor peptide complex**

The structure of the complex was solved by the molecular replacement method using the program MolRep (Vagin & Teplyakov, 2010), with SPD of human plasma kallikrein (PDB entry: 2ANY; Tang et al, 2005), which shows the highest sequence identity score (46.1%), as a search model. The model of SPD was manually fixed

with *COOT* (Emsley & Cowtan, 2004) and refined with Refmac5 (Murshudov et al, 2011). When the SPD of human MSPL was well refined, the interpretable electron density of the unmodeled region was observed, then the model of the LDLA and SRCR domains was manually built. The final model contained human MSPL, decanoyl–RVKR–cmk, four sugars, 80 ions, and 65 water molecules, with *R*-work and *R*-free values of 18.5% and 25.1%, respectively. The refinement statistics are summarized in Table S1. In the human MSPL–peptide inhibitor complex, some residues (N-terminal 3xFLAG-tag and His192, Gly324, Arg325, and C-terminal Gln564-Val 586) are missing because of disorder. All the structures in the figures were prepared using PyMOL v1.5.0 (http://www.pymol.org/). The MSPL–peptide inhibitor interfaces were analyzed using LigPlot+ (Laskowski & Swindells, 2011). Structure refinment statistics are summarized in Table S1.

**Homology modelling of human TMPRSS2**

The sequence alignment of the extracellular region of human MSPL and h uman TMPRSS2 was obtained using the BLAST web server (https://www.uniprot.org/blast/). The amino acid identity of extracellular regions between human MSPL and human TMPRSS2 was 39.8% with a score of 704, and E-value of 1.1 × 10$^{-86}$. The homology model of human TMPRSS2 was build using *MODELLER* (Šali & Blundell, 1993). Electrostatic surface potentials were calculated using the APBS server (http://server.poissonboltzmann.org/).

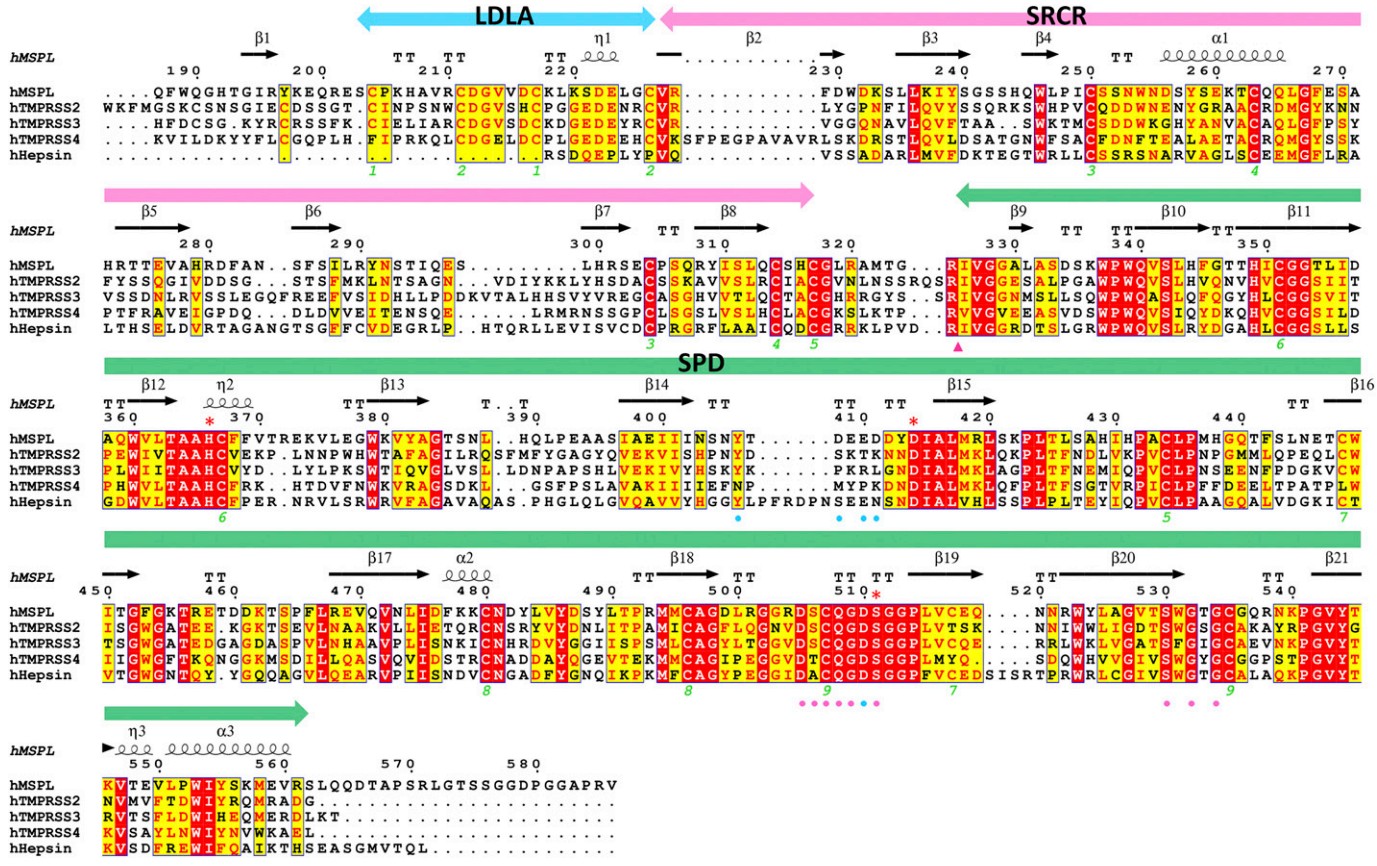

**Figure 7. Multiple sequence alignment of the human MSPL extracellular region with members of the hepsin/TMPRSS subfamily.**
The extracellular region of human MSPL (187–586), human TMPRSS2 (110–492), human TMPRSS3 (70–454), human TMPRSS4 (55–437), and human hepsin (50–417) aligned by *Clustal W* program (Thompson et al, 1994), followed by colouring with *ESPRIPT* (Gouet et al, 2003). Red asterisk indicates the catalytic triad. The amino acid sequences were obtained from UniProtKB with the id code of human MSPL (UniProt:Q9BYE2), human TMPRSS2 (UniProt:O15393), human TMPRSS3 (UniProt:P57727), human TMPRSS4 (UniProt:Q9NRS4), and human hepsin (UniProt:P05981). The secondary structure regions identified in MSPL are indicated. Identical residues are shown in white on red, whereas similar residues are shown in red. Pink arrowhead indicates the autocleavage site. Pink and cyan circles denote residues that interact with the P1 and P2 site of the decanoyl–RVKR–cmk inhibitor, respectively. Green numbers denote the disulfide pairing of human MSPL.

## Data Availability

The coordinates and structure factors of the human MSPL–decanoyl–RVKR–cmk inhibitor complex have been deposited to the RCSB (PDB entry: 6KD5). The homology model of human TMPRSS2 is available from supplementary materials (Supplemental Data 1).

## Supplementary Information

## Acknowledgements

We thank the beamline staff at the PF-AR and SPring-8 BL44XU for supporting data collection under the proposal number 2013G075 and 20156537, respectively. This work is supported by the Japan Society for the Promotion of Science (JSPS) KAKENHI grant number 15J40096 to A Ohno, 15K13747 & 19K05696 to N Maita, 15K09585 & 18K08453 to Y Okumura, and 18H04981 & 19H04054 to T Nikawa, and by AMED-CREST grant number JP19gm0810009h0104 to T Nikawa from Japan Agency for Medical Research and Development (AMED).

## Author Contributions

A Ohno: data curation, formal analysis, funding acquisition, validation, investigation, visualization, methodology, and writing—original draft.

N Maita: data curation, formal analysis, funding acquisition, validation, investigation, visualization, methodology, and writing—original draft.

T Tabata: resources, investigation, and methodology.

H Nagano: resources and investigation.

K Arita: methodology.

M Ariyoshi: validation, methodology, and writing—review and editing.

T Uchida: data curation, investigation, and methodology.

R Nakao: data curation, validation, and investigation.

A Ulla: formal analysis, investigation, and visualization.

K Sugiura: formal analysis and investigation.

K Kishimoto: conceptualization, data curation, and validation.

S Teshima-Kondo: conceptualization, data curation, and supervision.

Y Okumura: conceptualization, resources, formal analysis, supervision, funding acquisition, investigation, methodology, writing—original draft, and project administration.

T Nikawa: conceptualization, resources, supervision, project administration, and writing—review and editing.

**Conflict of Interest Statement**

The authors declare that they have no conflict of interest.

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
