## [Reviewer comments · Life Science Alliance]

Life Science Alliance

Crystal structure of inhibitor-bound human MSPL that can activate high pathogenic avian influenza

Ayako Ohno, NOBUO MAITA, Takanori Tabata, Hikaru Nagano, Kyohei Arita, Mariko Ariyoshi, Takayuki Uchida, Reiko Nakao, Anayt Ulla, Kosuke Sugiura, Koji Kishimoto, Shigetada Teshima-Kondo, Yuushi Okumura, and Takeshi Nikawa

DOI: <https://doi.org/10.26508/lsa.202000849>

Corresponding author(s): Yuushi Okumura, Sagami Women's University

Review Timeline:

Submission Date:	2020-07-13
Editorial Decision:	2020-08-24
Revision Received:	2020-11-09
Editorial Decision:	2020-12-07
Revision Received:	2021-02-24
Editorial Decision:	2021-03-09
Revision Received:	2021-03-12
Accepted:	2021-03-12

Scientific Editor: Shachi Bhatt

Transaction Report:

August 24, 2020

Re: Life Science Alliance manuscript #LSA-2020-00849

Prof. Yuushi Okumura
Sagami Women's University
Department of Nutrition and Health
2-1-1 Bunkyo, Minami
Sagamihara, Kanagawa 252-0383
Japan

Dear Dr. Okumura,

Thank you for submitting your manuscript entitled "Crystal structure of inhibitor-bound human MSPL that can activate high pathogenic avian influenza" to Life Science Alliance (LSA). The manuscript has been reviewed by the editors and outside referees (reviewer comments below). As you will see, the reviewers were enthusiastic about the study and its potential impact, but have raised a number of concerns that should be addressed prior to further consideration of the manuscript at LSA. Therefore, although we are unable to publish the current version of the manuscript, we would encourage you to submit a revised version that addresses both the referees' concerns, including the requests to improve figure quality and enlarged figures.

We would be happy to discuss the individual revision points further with you should this be helpful. The typical time frame for revisions is three months. Please note that papers are generally considered through only one revision cycle, so strong support from the referees on the revised version is needed for acceptance. When submitting the revision, please include a letter addressing the reviewers' comments point by point. While you are revising your manuscript, please also attend to the below editorial points to help expedite the publication of your manuscript. Please direct any editorial questions to the journal office.

Thank you for this interesting contribution to Life Science Alliance. We are looking forward to receiving your revised manuscript.

Sincerely,

Shachi Bhatt, Ph.D.
Executive Editor
Life Science Alliance
<https://www.life-science-alliance.org/>
Tweet to @SciBhatt @LSAjournal

B. MANUSCRIPT ORGANIZATION AND FORMATTING:

Reviewer #1 (Comments to the Authors (Required)):

The paper describes the crystal structure of inhibitor-bound human MSPL that activate high pathogenic avian influenza. MSPL consists of LDLA, SRCA, and SPD domains. The author solved the crystal structure of the extracellular region of human MSPL in complex with the furin inhibitor. The structure is similar to hepsin, but the arrangement of each domain is markedly different. The author compared the structure of MSPL and furin, which belongs to the different family protease trypsin-type and subtilisin-type. The furin inhibitor (decanoyl-RVKR-cmk) binds the MSPL active site despite the plat-form difference. Furthermore, the author did the homology modelling of TMPRSS2, which is an essential protease for SARS-CoV-2, based on the structure of the MSPL structure. The homology model is reasonable and explains how the inhibitor binds to TMPRSS2 and why TMPRSS2 prefers the monobasic inhibitor.

In the clear results and discussion, I highly recommend publishing in Life Science Alliance. I suggest a few minor points.

(1) Could you show the binding constant (or IC50) of the inhibitor to MSPL and furin?
From Fig.5, the inhibitor seems to bind to furin with its deep cleft.
However, MSPL does not have.
The each binding constant (IC50) reflects the structural feature?

(2) Please show the electron density of the inhibitor.

(3) Fig 6 B
Please use the same scaling in MSPL (small) and TMPRSS2.
It seems different.

(4) Author contribution
Please write all author contributions more clearly.

Reviewer #2 (Comments to the Authors (Required)):

At first, it was very interesting for me to read the submitted paper, which contains new and important results for all people working in the field. The authors describe for the first time the experimental structure of the complete extracellular region of MSPL (TMPRSS13), a member of the hepsin/TMPRSS subfamily of TTSPs. The authors have already released the pdb file of the determined MSPL structure in complex with an irreversible CMK inhibitor, therefore, I already could have a look on the complex. Based on the domain organization, MSPL is structurally closely related to TMPRSS2, a protease involved in the activation of the SARS-CoV-2 spike protein. Therefore, the authors have also provided a homology model of TMPRSS2.

However, according to my opinion the submission contains many unclear formulations and several errors. Therefore, the quality of the paper has to be considerably improved considering all my many comments listed below.

Page 2, Abstract

First sentence is wrongly written, of course not all enveloped viruses possess an hemagglutinin that is cleaved by these TTSPs, e.g., the gp160 of HIV is cleaved by furin-like PCs and the coronaviruses possess the so-called spike protein and not an HA, and certain H5N1 or H7N1 HPAIV HAs are cleaved by furin and not by TTSPs.

Correct this sentence to: Infection of certain influenza viruses is triggered when its ...

Line 4: In the consensus sequence (also in following sequences) I would always indicate the cleavage site after the P1 residue by a vertical arrow.

Line 5: In which paper the following statement was clearly demonstrated: while TMPRSS2 or -4 only cleaves monobasic motifs. The authors should be aware that the autocatalytic activation sequence of TMPRSS2 is RQSR|VGG (I have not checked this for TMPRSS4), but this statement has to be revised.

Line 7: replace "... of human MSPL in complex with the furin inhibitor..." by "...of human MSPL in complex with an irreversible substrate-analog inhibitor." (it was developed and described as a furin inhibitor, but it is a very nonspecific inhibitor that targets many other proteases as well)

Line 9: replace "... The furin inhibitor structure shows..." by "...The inhibitor structure shows..." (of course, it inhibits furin, but not a specific furin inhibitor, and it is sufficient to explain once in the abstract and once in the main text that this compound is a furin inhibitor)

Line 4 from bottom: Replace "... We also constructed a homology model of TMPRSS2, which is identified as an initiator of SARS-CoV-2 infection..." by "Based on the structure of MSPL, we also constructed a homology model of TMPRSS2, which is essential for the activation of the SARS-CoV-2 spike protein and infection."

Page 3:

Replace "The conserved Asp lies on the bottom of the S1 substrate-binding pocket." by "Like all other trypsin-like serine proteases, MSPL possesses a conserved Asp residue on the bottom of the S1 substrate-binding pocket, therefore, it accepts substrates with Arg or Lys in P1 position."

After spinesin I would write in brackets (TMPRSS5)

Replace: "... contains a low-density lipoprotein receptor A (LDLA) domain..." by "... contains an additional low-density lipoprotein receptor A (LDLA) domain...."

Page 4:

- The following statement should be written more precise: "It was recently reported that TMPRSS2, -4, and MSPL are involved in infections by influenza virus by cleaving the glycoprotein hemagglutinin (HA) on the influenza viral surface (4, 5, 12, 13, 14)." The HA cleavage mainly occurs in the secretory pathway (TGN, Golgi) long before it is incorporated into the viral surface during budding.

- Change "Specifically, HA is cleaved..." to "Specifically, certain HAs are cleaved..." (some HAs of HPAIN are cleaved by furin-like PCs)

- replace "viral-endosomal fusion" by "viral-endosomal membrane fusion"

Page 4:

As mentioned before related to the abstract, in all the provided consensus sequences I would suggest to always add a vertical arrow after the P1 residue.

What is the reason to mix round brackets and a slash in the consensus sequence, e.g. "Q(E)-T/X/-R"? Is there a difference in the meaning of a round bracket or slash? In case of P2 position very often also Ser occurs in the HA.

The following multibasic consensus sequence should be written as R-X-X/R-R|, otherwise the reader obtains the impression that the cleavage is between X/R.

Moreover, use a uniform style. Why using only once brackets and in other cases a diagonal slash, if different residues can occupy the same position, e.g. in case of (Q(E)-T/X/-R? That is confusing. I assume, it should be better written as Q/E-T/X-R|. This means, in P3 position Gln or Glu is accepted, in P2 position especially Thr but also most other residues.

The furin motive should be better written as R-X-X/R/K-R|, because furin also accepts very well Lys as P2 residue, but also non-basic residues, like Ala, as presently seen at the S1/S2 cleavage site of the CoV-2 spike protein.

line 4 from bottom: "However, it is not clear why only MSPL is able to recognize the multiple-basic-residue motif": You should be aware that matriptase also accepts very well substrates with basic residues in P1 and P4, sometimes in P3 (see: Takeuchi, T.; Harris, J. L.; Huang, W.; Yan, K. W.; Coughlin, S. R.; Craik, C. S. Cellular localization of membrane-type serine protease 1 and identification of protease-activated receptor-2 and single-chain urokinase-type plasminogen activator as substrates. *J Biol Chem* 2000, 275, 26333-26342).

Page 5:

replace "that mimics the substrate for the furin " with "...an irreversible substrate-analog inhibitor of furin" (but after this avoid to write always "furin inhibitor", simply write "inhibitor" or "used inhibitor" or "CMK-inhibitor".

related to the following sentence: "To date, only one structure of the extracellular region of hepsin has been reported among the hepsin/TMPRSS family of proteins (17)." It should be added that the SPD of enteropeptidase, which is an additional member of the TMPRSS2/hepsin family, has been determined, also in complex with an irreversible CMK inhibitor (see 1ekb.pdb). Of course, enteropeptidase contains many additional domains, but for the design of synthetic inhibitors, the knowledge of the 3D structure of the SPD is usually sufficient.

replace "...in complex with the decanoyl-RVKR-cmk peptide..." by "... in complex with the irreversible inhibitor decanoyl-RVKR-cmk".

- I would appreciate, if the authors can provide the pdb file of their complete TMPRSS2 model with all domains as supporting information, so interested readers can have an own look on the established model.

Figure 1:

I would appreciate, if Figures 1B and C can be considerably enlarged, at least twofold. Perhaps the residue numbers on the domains in panel 1A could be added. I would also appreciate, if the disulfide bonds between the different domains and within the domains could be added. There is sufficient space in electronic journals to enlarge such Figures. As mentioned before, in the whole manuscript at most places (e.g., also in caption of Figure 1B, 1C) I would replace the words "furin inhibitor" by "irreversible inhibitor" or "used inhibitor" (obviously, it is also a TMPRSS13 inhibitor and it inhibits also other trypsin-like serine proteases).

I am also confused about the statement on page 6 (five lines from bottom) that the LDLA domain is 24 residues in length, but in the caption for Fig. 1A the authors describe the truncated LDLA domain as the sequence 204-220, which correspond to 17 residues, if I have this correctly counted. In contrast, at the top of page 6 the authors describe in an additional sentence that the LDLA domain comprises residues 298-221, this would be 24 residues. Such inconsistencies make it a little bit frustrating for a reviewer. I simply expect an accurate consistent description.

Caption to Figure 1B: For all residues of the SPD domain always provide also the number based on the chymotrypsin numbering in round brackets after the full-length numbering. For instance also for Asn400 (not for Asn250 which is outside of the SPD). This is much more convenient for the reader, because nobody knows the individual full length numbering of these TTSPs, but many people know the chymotrypsin numbering. In panel 1C the catalytic triad can be seen, which is Ser506(195), His361(57) and Asp409(102). Based on the uniprot entry Q9BYE2 I had problems to find the residues of the catalytic triad, based on the provided residue numbers, but based on the caption of Figure 4 the authors have also used this uniprot number. The active site Ser is #511 and not 506,

the Asp102 should be #414 and not #409, the His57 should be number 366 and not 361. Is there any good explanation for the 5 residue shift used by the authors compared with the uniprot entry Q9BYE2? I do not understand this discrepancy.

I personally would even prefer to write first the chymotrypsin numbering for the SPD, followed by the full-length numbering in brackets, if possible. But the opposite is also ok. Again, this comment only refers to the numbering of the SPD, which always starts at residue 16 based on the chymotrypsin/chymotrypsinogen numbering.

Page 6:

Also in the text: "Glycans attached to residues Asn250 and Asn400 were observed". In this case add the chymotrypsin numbering for residue Asn400. However, the authors should check this number. Based on the uniprot entry for MSPL (UniProtKB - Q9BYE2 (TMPD_HUMAN) residue 400 is Ile, two Asn residues exist at positions 403 and 405. Please check and correct this, if necessary. The authors should also check all other residue numbers they have used for correctness.

The hMSPL is activated by hydrolytic cleavage at Arg320-Ile321 and residues: replace Ile321 by Ile321(16), because this is residue 16 which always interacts with Asp194 beside the active site Ser195 in case of all serine proteases of the S1A subfamily. Here again, there is a discrepancy, in the uniprot entry the SPD starts at #326 and not at #321! Please clarify these differences.

"321-581 region are converted to the mature SPD", add the number of residue321 and 581 based on chymotrypsin in brackets.

Replace "Ile321 is located in a pocket where the N atom interacts with Asp505 (Fig. S1A)." by "Ile321(16) is located in a pocket where the N atom interacts with the side chain of Asp505(194) (Fig. S1A)." This is the common and well activation mechanism for all trypsin-like serine proteases, a reference should be added.

Figure 2: Again, all Figure have to be considerably enlarged at least two-fold, there is no reason to provide such tiny Figures! Panel C: add the information regarding the used colors for the domains, I assume green = SPD, blue LDLA, magenta = SRCR.

"Accordingly, a short parallel β -sheet between the N-terminal segment and the SPD domain was observed in MSPL, whereas the C-terminal end of hepsin forms an antiparallel β -sheet (Fig. 2A)": Unfortunately, I cannot see this interaction in this small panel 2A, may be you can add an inset for this panel which shows the described interaction.

Page 7:

Hence, both MSPL and hepsin may bind substrate in close proximity to the transmembrane region: Replace this by: "Hence, both MSPL and hepsin may bind their substrates in close proximity to the transmembrane region."

Third line from bottom: After Arg 556 add the chymotrypsin numbering.

Page 8:

Change the title of the paragraph to: Interaction of the inhibitor decanoyl-RVKR-cmk in the active site of MSPL

replace "S1 family" by "S1A family of serine protease" (see the nomenclature given in the Merops

database)

In the activated MSPL, Ile321 at the cleavage site forms a salt bridge with the conserved Asp505 residue located immediately prior to the catalytic Ser506 residue (Fig. S1A).: change to: "In the activated MSPL, Ile321(16) at the cleavage site forms a salt bridge with the conserved Asp505(194) residue located immediately prior to the catalytic Ser506(195) residue (Fig. S1A)."

Of course, somewhere at the beginning of the manuscript the authors should provide a hint that always the chymotrypsin numbering is used in round brackets after the full-length number for residues of the SPD.

Change "This interaction might be generated by the activating cleavage." To "This interaction is generated by the activating cleavage." Please provide a reference for this common mechanism, which was several times nicely described by Wolfram Bode (e.g., for tPA, although there the zymogen has relatively high activity: Stubbs, M. T.; Renatus, M.; Bode, W. An active zymogen: unravelling the mystery of tissue-type plasminogen activator. *Biol Chem* 1998, 379, 95-103.) But this paper nicely explains the activation mechanism.

"This salt bridge was also observed in other proteases such as plasma kallikrein (20)(PDB entry: 1Z8G) and hepsin.": This you will find for each of the 60/70 trypsin-like serine protease, a little exception is tPA, where the zymogen already possesses considerable activity.

"A furin inhibitor peptide binds to the SPD of MSPL with P1-Arg, P2-Lys, C-terminal cmk (chloromethylketone; an active site-direct group) and N-terminal decanoyl group (Fig. 1C, 3)." change to "The chloromethyl group of the inhibitor irreversibly alkylates His57 of the SPD of MSPL, in addition, an hemiketal is formed to the active site Ser195. In addition, several interactions via the P1-Arg and P2-Lys side chains are formed (Fig. 1C, 3).

"to the backbone amides of the oxyanion hole (Gly504 and Ser506)" replace by ".....Gly504(193) and Ser506(195)"

"as a hydrogen bond with the side chain of Ser501 and the backbone carbonyl of Gly5292 should be replaced by "as a hydrogen bond with the side chain of Ser501(190) and the backbone carbonyl of Gly529(219)." The authors should check this, but I think these are the numbers of the SPD.

Write: "Asp500(189) is located in the bottom of S1 pocket."

There seem to exist significant errors related to the numbering and text in the following sentence: "...Lys interacts with residues at the so-called 99-loop (99 comes from the chymotrypsinogen numbering!) that contains the catalytic residue Asp409" Therefore, I always suggest to use the SPD numbering, because otherwise the name 99-loop would not make sense if it is located close to residue 409! However, I checked the uniprot entry mentioned before, there residue 409 is a Glu and not Asp! The catalytic Asp102 is number 414 on the full-length numbering in the file I have used. Moreover, I would not say that Asp 102 belongs to the 99 loop, although it is close to residue 99. The whole residue numbers for all comments made in the text and the related sentences have to be revised and corrected, if necessary. Maybe, I am wrong, but I simply had a look at the uniprot entry. Interestingly, before Asp414(102) there are five (!) additional acidic residues DEEDD (sequence 408-412 corresponding to 96-100 based on chymotrypsinogen).

Page 9:

"with the backbones of Asp403 and Glu405, the side chains of Tyr401 and Asp406...": also here,

the correct chymotrypsin numbering should be added. Again, in the uniprot entry Q9BYE2[1 - 586] I could not find Asp403 and Asp405 with these numbers, at position 401 I have found this sequence: I401-INSNYTDEE410

chains of Asp406 and the catalytic Asp409 residue: same, again is 409 really the catalytic Asp102, which residue is 406?

Based on all these trouble with the residue numbers (may be, I am wrong), I suggest to revise Figure 4 and add as first entry the sequence of chymotrypsin (this starts at residue 1), in this case the SPD of all other proteases start at residue 16. By doing this, the authors can easily deduce the appropriate residue numbers.

The chymotrypsin numbering should be also added in brackets for the residues of the SPD in the Ligplot (Figure 3).

Two mistakes in the following statement: "... there is no hydrogen bond between the side chains of P3-Val/P4-Lys and the MSPL." Sentence makes no sense and is wrong. It is clear that no H-bond can be formed from a Val side chain! Moreover, the CMK inhibitor possesses an Arg in P4 and not Lys!

Replace Gly527 by Gly527(216)

"The side chain of P3-Val makes van der Waals interactions with Trp526 and Gly527". This is very unusual, but I have seen it in the pdb file. Normally, a P3 side chain of a residue in L-configuration goes into the solvent. Moreover, this is a very unusual backbone conformation, in which the P3 NH goes up into the solvent. In all other structures which I know for trypsin-like serine proteases in complex with substrate analog inhibitors this NH goes down to the carbonyl of Gly216, it means, the P3 backbone forms a short antiparallel beta-sheet interaction with Gly216. This very artificial binding mode is probably caused by the presence of the artificial sulfate ion.

During looking on the pdf file I have also realized an unusual (most-likely wrong) torsion angle for the amide bond between the decanoyl residue and the P4 Arg backbone NH, which looks like approximately 90 degree. I assume, it should be 180 degree (trans amide bond). The authors should check, if there is an error in their structure and revise their pdb file.

Figure 4, showing the sequence alignment based on the full-length numbering should be also enlarged. Moreover, it would be great to add an additional panel 4B, showing only the alignment of the SPD with chymotrypsin as first line. This would provide the correct numbers for the SPD based on chymotrypsin.

Figure 5: Figure is also very tiny and should be enlarged. Moreover, I strongly suggest to show all structures of active site inhibitors for trypsin-like serine proteases or furin-like PCs in the so-called standard orientation, suggested always by W. Bode several decades ago (as example see the 2003 Henrich paper showing the crystal structure of the used inhibitor in complex with furin (PDB: 1P8J, Figure 4, on the right we have the P1 residue, than the peptide backbone goes from the right to the left side...)

Page 10:

"As a consequence, the P1, P2, and P4 site contacts with furin, whereas the P3 site is directed away from it": A hint should be given that in case of furin the P3 Val backbone makes an important

antiparallel beta-sheet interaction with furin residue Gly255 (this is the equivalent residue to Gly216 in proteases of the S1A family), whereas in the MSPL structure the P3 residue makes only one backbone interaction, probably caused by the presence of the artificial sulfate ion.

"To date, the structure of TMPRSS2 has not been reported." change to "To date, the experimental structure of TMPRSS2 has not been reported." At least for the SPD, there exist several homology models, but probably not for the full-length extracellular domain.

Eight out of nine disulfide bonds are conserved (Fig. 4): It is very hard to see them in this very small Figure 4 showing the alignment. Perhaps, an additional schematic structure for TMPRSS2 could be added (like in Figure 1A, e.g., in the supporting info) in direct comparison to MSPL showing the disulfide bonds (then the disulfide bonds have not to be shown in Figure 1A).

Furthermore, Glu404, an important residue for P2-Lys recognition in MSPL, is replaced by Lys225 in TMPRSS2 (Fig. 4, 6B): I wonder about these many discrepancies in the residue numbering compared with the uniprot entry for human MSPL. In their crystal structure there is absolute no interaction between Glu404 and the P2 Lys side chain, there exists an interaction between the Glu405 carbonyl oxygen! This error is independent from any confusion regarding the residue numbering. I would appreciate, if the responsible author of this paper would have an own look on the structure!

I am disappointed about all these mistakes, because it makes a lot of work for a reviewer. Originally, I assume, the authors speak from residue 99 at the tip of the 99 hairpin loop, correct? I know that this is Lys99 in case of TMPRSS2. Please use the chymotrypsin numbering!

Figure 6: The Figure has to be improved and enlarged (all panels). The inhibitor should be added to panel A to see, if this is placed close to the region in the red rectangle, where differences between TMPRSS2 and MSPL exist, or if it bound far away (what should be the case).

"As mentioned earlier, this substitution leads to a preference for the monobasic target of TMPRSS2. In fact, the S1/S2 cleavage site of SARS-CoV-2 spike protein is reported as P2-Ala instead of a basic residue (25, 26, 27). In summary, our homology model reflects the features of TMPRSS2 target peptide recognition.": This whole paragraph gives a very wrong impression that the S1/2 site in S of SARS-CoV-2 with Ala in P2 position is cleaved by TMPRSS2, which is not the case. This site is cleaved/activated by furin! TMPRSS2 is supposed to cleave at the S2 site of SARS-CoV-2, and this sequence is PSKPSKR|SFIEDL, it means it also has a basic residue in P2 position (see the recent paper, Bestle et al. 2020 in Life Sci Alliance. 2020 Jul 23;3(9):e202000786.).

Page 11, conclusion section :

"HAPI virus" correct to "HPAI virus" or "HPAIV" (it comes from highly pathogenic avian influenza virus).

"MSPL also contributes to the cleavage and activation of severe acute respiratory syndrome coronavirus (SARS-CoV) Middle East respiratory syndrome coronavirus (MERS-CoV) spike proteins": A comment/reference should be given, at which site MSPL contributes to the S activation of these additional Corona viruses. There are always two cleavage sites in the S protein of CoV.

Dear Editor,

Thank you very much for your e-mail dated Aug 24, 2020, concerning our manuscript (number LSA-2020-00849). Please find attached the manuscript that has been revised in accordance with the suggestions of the reviewers.

First of all, we would like to express our sincere thanks to the editor and reviewers for carefully evaluating the manuscript and providing helpful comments. We have read the comments carefully and have made revisions and improvements to the main text and figures. In particular, we have rewritten the "Abstract and Introduction" sections in accordance with the reviewer's instructions.

A major change in our Results is that the P3-P4 region of the inhibitor peptide is likely to be an artifact due to crystal packing. As such, we discuss the interaction of the inhibitor by modeling based on the hepsin:peptide complex structure. This model implies that P4-Arg can be recognized in the 99-loop amino acid cluster, which is specific to MSPL, and that MSPL can recognize the target sequence containing Lys/Arg at position P4.

We have also corrected the claim made in the previous manuscript that TMPRSS2 would cut S1/S2 in SARS-CoV-2. We accept that this was wrong; rather TMPRSS2 cuts the site at S2'.

We have revisited the TMPRSS2 homology model. The sequence of S2' sites in SARS-CoV-2 is KPSKR↓SFIE, which is P2-Lys. The S2 sites are sufficiently negative in the electrostatic surface potential and therefore the influence of Lys225 in TMPRSS2 seems to be minimal in terms of P2-Lys binding. By contrast, the S1' site (P1' binding site) of TMPRSS2 comprises a wide bowl shape and can bind target peptides of various structures (i.e. alpha-helix) downstream of P1'. The recently reported cryo-EM structure of SARS-CoV-2 shows that an α -helix structure was taken from P1'-Ser. Therefore, it is conceivable that the TMPRSS2 would need to cleave the SARS-CoV-2 S2' site with the α -helix structure intact. Indeed, this is consistent with the wider S1' site of TMPRSS2. Based on these considerations, we have substantially rewritten the section of text concerning TMPRSS2. Along with the revisions, a total 60 references are cited.

Major Revisions

1. A conformational error on decanoyl-RVKR-cmk structure.
According to the reviewer's comment concerning modelling at the P4-Arg-dec linkage, we have re-examined the structural refinement and fixed the anomalies. The Ramachandran plot analysis of the corrected structure of P4-Arg is within the favored region (attached figure). The structure will be updated in PDB, and Table S1 is also updated accordingly.
2. An artificial inhibitor conformation.

In accordance with the reviewer's comment, we revisited the conformation of the inhibitor. Upon further examination, we concluded that conformation of the P3-P4 moiety is likely to be an artifact due to crystal packing (new figure; Fig 4). Next, we modelled the putative RVKR peptide using the hepsin: ace-KQLR-cmk structure (PDB: 1Z8G) as a template and considered the P4-Arg interaction. We added the putative model in Fig S2. Please also refer to the section entitled "Interaction of the inhibitor decanoyl-RVKR-cmk in the active site of MSPL" in the revised manuscript.

3. Figure addition

New figures describing the putative RVKR peptide were added in Figure 4. Accordingly, old figure 4, 5, and 6 were changed to figure 5, 6, and 7, respectively.

4. TMPRSS2 function on SARS-CoV-2 activation.

In accordance with the reviewer's comment, we revisited the target sequence specificity of TMPRSS2. Although TMPRSS2 mainly cleaves the monobasic sequence (Böttcher-Friebertshäuser et al, Pathog Dis 2013), it also cleaves di- or tri-basic sequences (Baron et al, JVI, 2013). Furthermore, recent studies have established that the SARS-CoV-2 S-protein is activated by furin mediated cleavage at the S1/S2 site, followed by TMPRSS2 mediated cleavage at the S2' site (Hoffman et al, Mol Cell, 2020; Bestle et al, LSA, 2020). The S2' site sequence of SARS-CoV-2 is KPSKR↓SFI. Thus, we retract our conclusion that Lys225 reduces the affinity of TMPRSS2 for basic residues at the P2 site. As shown in Fig 7B, the electrostatic surface potential around the P2-Lys binding site is negatively charged, suggesting that Lys/Arg at the P2 site is acceptable. We focused on the characteristic 3D structure of the S-protein S2' site, which forms a loop-helix structure. The alpha-helix starts from the P1'-Ser (PDB: 6XR8; new Figure S2), which may require a wide binding cleft at the P1' position for TMPRSS2. Indeed, these conclusions are consistent with our homology model. We added a discussion in the "*Homology model analysis of TMPRSS2*" section.

5. Ki values of the inhibitor.

In accordance with the reviewer's comment, we added Ki values of decanoyl-RVKR-cmk to MSPL (2.9 nM) and furin (0.6 nM) from previous studies. Please refer to the section entitled "*Comparison of the binding mechanisms of decanoyl-RVKR-cmk inhibitor to human MSPL and furin*".

6. Figure quality.

All figures were updated to high resolution (450dpi) for publication.

7. Chymotrypsin numbering

To clarify the discussion of chymotrypsin as reference for the S1A superfamily, we added the sequence alignment of bovine chymotrypsin and SPD of human MSPL in Fig 3. In addition, we now describe chymotrypsin numbering in parentheses.

Minor Revisions

8. Residue numbering
For clarity, the residue numbering of MSPL refers to UniProt Q9BYE2.
9. Homology model deposition
The Tmprss2 model that we build is deposited to supplemental materials.
10. Comparison of Tmprss2/hepsin subfamily
For clarity, a schematic drawing of human Tmprss2 and hepsin is added to Fig. 1A. The disulfide bond and glycosylation sites are also indicated.
11. Electron density
In accordance with the reviewer's comment, we added the electron density of the inhibitor in Fig 1C.
12. In accordance with the reviewer's comment, the close-up view of the N-terminal beta-strand was added in Fig 2A.
13. To clarify the origin of proteins, we describe the prefix of species as human MSPL, and hTmprss2.
14. We highlight the cleavage position in target sequences using an arrow (↓) throughout the manuscript.

Response to Reviewer #1 comments:

(1) Could you show the binding constant (or IC50) of the inhibitor to MSPL and furin?

From Fig.5, the inhibitor seems to bind to furin with its deep cleft. However, MSPL does not have. The each binding constant (IC50) reflects the structural feature?

The K_i values of decanoyl-RVKR-cmk for furin and MSPL are reported previously. K_i for furin is 0.6 nM (Jean et al, PNAS, 1998), and MSPL is 2.9 nM (Okumura et al, J Biol Chem, 2010). We added the K_i values in the section entitled "*Comparison of the binding mechanisms of decanoyl-RVKR-cmk inhibitor to human MSPL and furin*".

(2) Please show the electron density of the inhibitor.

We added the electron density (2Fo-Fc) of the inhibitor in Fig 1C.

(3) Fig 6 B

Please use the same scaling in MSPL(small) and Tmprss2.

It seems different.

We have redrawn Fig 6B to adjust the size of Tmprss2.

(4) Author contribution

Please write all author contributions more clearly.

We describe all author's contributions according to the journal's format.

Response to the Reviewer #2

Page 2, Abstract

(1) First sentence is wrongly written, of course not all enveloped viruses possess an hemagglutinin that is cleaved by these TTSPs, e.g., the gp160 of HIV is cleaved by furin-like PCs and the coronaviruses possess the so-called spike protein and not an HA, and certain H5N1 or H7N1 HPAIV HAs are cleaved by furin and not by TTSPs.

Correct this sentence to: Infection of certain influenza viruses is triggered when its

We have corrected the sentence as suggested in the reviewer's comment (refer to the Abstract section).

(2) Line 4: In the consensus sequence (also in following sequences) I would always indicate the cleavage site after the P1 residue by a vertical arrow.

We highlight the cleavage site with vertical arrows throughout the manuscript.

(3) Line 5: In which paper the following statement was clearly demonstrated: while TMPRSS2 or -4 only cleaves monobasic motifs. The authors should be aware that the autocatalytic activation sequence of TMPRSS2 is RQSR|IVGG (I have not checked this for TMPRSS4), but this statement has to be revised.

The sentence "TMPRSS2 or -4 only cleaves monobasic motifs." was incorrect. The monobasic motif can be cleaved by several TTSPs including TMPRSS2 -4. This does not mean that TMPRSS2 and 4 cannot cleave other monobasic motifs. It is reported that TMPRSS2 cleaves autoactivate motif (RQSR↓) and SARS-CoV-2 S-protein (KPSKR↓), and TMPRSS4(CAP2) cleaves HA0 (IQSR↓), and gamma-ENaC subunit (RKRR↓). Therefore, we changed the sentence "**HA with a monobasic motif is cleaved by chymotrypsin-like proteases, including TMPRSS2 and HAT, while the multibasic motif found in high pathogenicity avian influenza HA are cleaved by furin, PC5/6, and MSPL.**" In abstract.

(4) Line 7: replace ".... of human MSPL in complex with the furin inhibitor...." by "....of human MSPL in complex with an irreversible substrate-analog inhibitor." (it was developed and described as a furin inhibitor, but it is a very nonspecific inhibitor that targets many other proteases as well)

In accordance with the Reviewer's comment, we replace the term "furin inhibitor" with "irreversible substrate-analog inhibitor" or "decanoyl-RVKR-cmk" throughout the manuscript.

(5) Line 9: replace "... The furin inhibitor structure shows..." by "...The inhibitor structure shows..." (of course, it inhibits furin, but not a specific furin inhibitor, and it is sufficient to explain once in the abstract and once in the main text that this compound is a furin inhibitor)

According to the comment, we replaced the sentence with "The inhibitor structure and its putative model show" on page 2, line 11-12. The underlined words were added because the P3-P4 region of the inhibitor had the wrong conformation, and the predicted model was used to discuss the target preference of MSPL.

(6) Line 4 from bottom: Replace "... We also constructed a homology model of TMPRSS2, which is identified as an initiator of SARS-CoV-2 infection..." by "Based on the structure of MSPL, we also constructed a homology model of TMPRSS2, which is essential for the activation of the SARS-CoV-2 spike protein and infection."

We replaced the sentence as suggested by the reviewer's comment (refer to at page 2, line 14-16).

Page 3:

(7) Replace "The conserved Asp lies on the bottom of the S1 substrate-binding pocket." by "Like all other trypsin-like serine proteases, MSPL possesses a conserved Asp residue on the bottom of the S1 substrate-binding pocket, therefore, it accepts substrates with Arg or Lys in P1 position."

We replaced the sentence as suggested by the reviewer's comment (refer to page 3, line 18-20).

(8) After spinesin I would write in brackets (TMPRSS5)

We added (TMPRSS5) after spinesin at the bottom of page 3.

(9) Replace: "... contains a low-density lipoprotein receptor A (LDLA) domain..." by "... contains an additional low-density lipoprotein receptor A (LDLA) domain...."

We replaced the sentence as suggested by the reviewer's comment (refer to page 4, line 2).

Page 4:

(10) - The following statement should be written more precise: "It was recently reported that TMPRSS2, -4, and MSPL are involved in infections by influenza virus by cleaving the glycoprotein hemagglutinin (HA) on the influenza viral surface (4, 5, 12, 13, 14)." The HA cleavage mainly occurs in the secretory pathway (TGN, Golgi) long before it is incorporated into the viral surface during budding.

According to the comment, we rewrote the sentence as "Previous studies show that TMPRSS2, -4, DESC1, HAT, and MSPL activate the influenza virus by cleaving HA0 (Böttcher et al, 2006; Chaipan et al, 2009; Okumura et al, 2010; Antalis et al, 2011; Ohler & Becker-Pauly, 2012;

Böttcher-Friebertshäuser et al, 2013; Zmora et al, 2014; Böttcher-Friebertshäuser, 2018). A newly synthesized HA is cleaved during its transport to the plasma membrane in the trans-Golgi network by furin or TMPRSS2, whereas HAT cleaves it at the cell surface during viral budding (Böttcher-Friebertshäuser, 2018).". Two additional review papers have been included in the references.

(11) - Change "Specifically, HA is cleaved..." to "Specifically, certain HAs are cleaved..." (some HAs of HPAIV are cleaved by furin-like PCs)

We rewrote the Introduction, and replaced the sentences "Specifically, HA is cleaved into HA1 and HA2 subunits by TMPRSS2, -4, and MSPL. Proteolytic cleavage of HA is essential for influenza virus infection, where HA1 mediates host cell binding as well as initiation of endocytosis and HA2 controls viral-endosomal fusion (15)." by "**The influenza viral hemagglutinin (HA0) is cleaved by various host proteases, and divided into HA1 and HA2 subunits, where HA1 mediates host cell binding as well as the initiation of endocytosis and HA2 controls viral-endosomal membrane fusion (Hamilton et al, 2012).**" because HA is also cleaved by many proteases (furin, HAT, PC etc). Refer to page 4, line 16–19.

(12) - replace "viral-endosomal fusion" by "viral-endosomal membrane fusion"

As suggested, we replaced "viral-endosomal fusion" with "viral-endosomal membrane fusion" on page 4, line 18-19.

Page 4:

(13) As mentioned before related to the abstract, in all the provided consensus sequences I would suggest to always add an vertical arrow after the P1 residue.

According to the comment, we include an arrow (↓) to highlight the cleavage position within the target sequences throughout the manuscript.

(14) What is the reason to mix round brackets and a slash in the consensus sequence, e.g. "Q(E)-T/X/-R"? Is there a difference in the meaning of a round bracket or slash? In case of P2 position very often also Ser occurs in the HA.

Apologies for the confusion. Q(E) and Q/E refers to "Gln or Glu". In the revised manuscript we unify the description using a slash.

(15) The following multibasic consensus sequence should be written as R-X-X/R-R|, otherwise the reader obtains the impression that the cleavage is between X/R.

Moreover, use a uniform style. Why using only once brackets and in other cases a diagonal slash, if different residues can occupy the same position, e.g. in case of (Q(E)-T/X/-R? That is confusing. I assume, it should be better written as Q/E-T/X-R|. This means, in P3 position Gln or Glu is

accepted, in P2 position especially Thr but also most other residues.

To avoid any confusion, we use an arrow (↓) to highlight the cleavage position within target sequences throughout the manuscript.

(16) The furin motive should be better written as R-X-X/R/K-R], because furin also accepts very well Lys as P2 residue, but also non-basic residues, like Ala, as presently seen at the S1/S2 cleavage site of the CoV-2 spike protein.

We have made extensive changes to the Introduction section. Rather than describe the furin consensus sequence, we simply indicate furin prefers the multibasic motif found in HPAIV.

(17) line 4 from bottom: "However, it is not clear why only MSPL is able to recognize the multiple-basic-residue motif": You should be aware that matriptase also accepts very well substrates with basic residues in P1 and P4, sometimes in P3 (see: Takeuchi, T.; Harris, J. L.; Huang, W.; Yan, K. W.; Coughlin, S. R.; Craik, C. S. Cellular localization of membrane-type serine protease 1 and identification of protease-activated receptor-2 and single-chain urokinase-type plasminogen activator as substrates. J Biol Chem 2000, 275, 26333-26342).

Because the multibasic motif is cleaved by furin/PC and matriptase, we revisited the target sequence of MSPL. We now focus on P4-Lys, which is not cleaved by furin/PC. This topic is discussed in the section entitled "Comparison of the binding mechanisms of decanoyl-RVCR-cmk inhibitor to human MSPL and furin".

Page 5:

(18) replace "that mimics the substrate for the furin " with ...,an irreversible substrate-analog inhibitor of furin" (but after this avoid to write always "furin inhibitor", simply write "inhibitor" or "used inhibitor" or "CMK-inhibitor".

We rewrote the Introduction section. The sentence "A previous study showed that the enzyme activity of MSPL was inhibited by decanoyl-RVCR-cmk that mimics the substrate for the furin (5)." has been removed. However, the inhibitor is described as "decanoyl-RVCR-cmk" throughout the modified manuscript.

(19) related to the following sentence: "To date, only one structure of the extracellular region of hepsin has been reported among the hepsin/TMPRSS family of proteins (17)." It should be added that the SPD of enteropeptidase, which is an additional member of the TMPRSS2/hepsin family , has be determined, also in complex with an irreversible CMK inhibitor (see 1ekb.pdb). Of course, enteropeptidase contains many additional domains, but for the design of synthetic inhibitors, the knowledge of the 3D structure of the SPD is usually sufficient.

In accordance with this comment, we rewrote the sentence as "To date, the extracellular region of human hepsin (Somoza et al, 2003; Herter et al, 2005) and SPD of enteropeptidase (Lu et al,

1999; Simeonov et al, 2012) are the only structures reported among the hepsin/TMPRSS family.” on page 5, line 13-15.

(20) replace "...in complex with the decanoyl-RVKR-cmk peptide..." by "... in complex with the irreversible inhibitor decanoyl-RVKR-cmk".

In accordance with this comment, we replaced the sentence on page 5, line 21-22.

(21) - I would appreciate, if the authors can provide the pdb file of the their complete TMPRSS2 model wit all domains as supporting information, so interested readers can have an own look on the established model.

We have deposited the PDB file of the TMPRSS2 model as supplemental data.

Figure 1:

(22) I would appreciate, if Figures 1B and C can be considerably enlarged, at least twofold. Perhaps the residue numbers on the domains in panel 1A could be added. I would also appreciate, if the disulfide bonds between the different domains and within the domains could be added. There is sufficient space in electronic journals to enlarge such Figures. As mentioned before, in the whole manuscript at most places (e.g., also in caption of Figure 1B, 1C) I would replace the words "furin inhibitor" by "irreversible inhibitor" or "used inhibitor" (obviously, it is also a TMPRSS13 inhibitor and it inhibits also other trypsin-like serine proteases).

According to the comment, we redrew Figure 1 as follows;

- 1) The Figure has improved quality (450dpi).
- 2) The residue numbers on the domains in panel 1A have been added.
- 3) Disulfide bonds (and N-glycans) are also shown.
- 4) “Furin inhibitor” is replaced by “decanoyl-RVKR-cmk”.
- 5) A schematic illustration of TMPRSS2 and hepsin has been added.
- 6) The electron density of decanoyl-RVKR-cmk is shown in Fig 1C.

(23) I am also confused about the statement on page 6 (five lines from bottom) that the LDLA domain is 24 residues in length, but in the caption for Fig. 1A the authors describe the truncated LDLA domain as the sequence 204-220, which correspond to 17 residues, if I have this correctly counted. In contrast, at the top of page 6 the authors describe in an additional sentence that the LDLA domain comprises residues 298-221, this would be 24 residues. Such inconsistencies make it a little bit frustrating for a reviewer. I simply expect an accurate consistent description.

We made a mistake in the Figure 1A legend. The LDLA domain region 203-226, 24-aa long is correct. We have corrected both the legend and figure.

(24) Caption to Figure 1B: For all residues of the SPD domain always provide also the number based on the chymotrypsin numbering in round brackets after the full-length numbering. For

instance also for Asn400 (not for Asn250 which is outside of the SPD). This is much more convenient for the reader, because nobody knows the individual full length numbering of these TTSPs, but many people know the chymotrypsinogen numbering. In panel 1C the catalytic triad can be seen, which is Ser506(195), His361(57) and Asp409(102). Based on the uniprot entry Q9BYE2 I had problems to find the residues of the catalytic triad, based on the provided residue numbers, but based on the caption of Figure 4 the authors have also used this uniprot number. The active site Ser is #511 and not 506, the Asp102 should be #414 and not #409, the His57 should be number 366 and not 361. Is there any good explanation for the 5 residue shift used by the authors compared with the uniprot entry Q9BYE2? I do not understand this discrepancy.

For clarity, the human MSPL numbering corresponds to UniProt# Q9BYE2. Also, the chymotrypsin numbering and amino acids are added in parentheses after the MSPL SPD residues.

(25) I personally would even prefer to write first the chymotrypsin numbering for the SPD, followed by the full-length numbering in brackets, if possible. But the opposite is also ok. Again, this comment only refers to the numbering of the SPD, which always starts at residue 16 based on the chymotrypsin/chymotrypsinogen numbering.

The amino acids of chymotrypsin and numbering are added in parentheses after the MSPL SPD residues.

Page 6:

(26) Also in the text: "Glycans attached to residues Asn250 and Asn400 were observed". In this case add the chymotrypsin numbering for residue Asn400. However, the authors should check this number. Based on the uniprot entry for MSPL (UniProtKB - Q9BYE2 (TMPD_HUMAN) residue 400 is Ile, two Asn residues exist at positions 403 and 405. Please check and correct this, if necessary. The authors should also check all other residue numbers they have used for correctness.

For clarity, the human MSPL numbering corresponds to UniProt# Q9BYE2. Also, the chymotrypsin numbering and amino acids are added in parentheses after the MSPL SPD residues.

(27) The hMSPL is activated by hydrolytic cleavage at Arg320-Ile321 and residues: replace Ile321 by Ile321(16), because this is residue 16 which always interacts with Asp194 beside the active site Ser195 in case of all serine proteases of the S1A subfamily. Here again, there is a discrepancy, in the uniprot entry the SPD starts at #326 and not at #321! Please clarify these differences.

"321-581 region are converted to the mature SPD", add the number of residue321 and 581 based on chymotrypsin in brackets.

For clarity, the human MSPL numbering corresponds to UniProt# Q9BYE2. Also, the chymotrypsin numbering and amino acids are added in parentheses after the MSPL SPD residues.

(28) Replace "Ile321 is located in a pocket where the N atom interacts with Asp505 (Fig. S1A)." by "Ile321(16) is located in a pocket where the N atom interacts with the side chain of Asp505(194) (Fig. S1A)." This is the common and well activation mechanism for all trypsin-like serine proteases, a reference should be added.

We replaced "Ile321" by "Ile326(Ile16)", and added the sentence and reference as follows: "This "Ile16-Asp194 salt-bridge" is a common feature among the trypsin-like proteases (Halfon & Craik, 1998)." Refer to page 9, line 19-20.

(29) Figure 2: Aagain, all Figure have to be considerably enlarged at least two-fold, there is no reason to provide such tiny Figures! Panel C: add the information regarding the used colors for the domains, I assume green = SPD, blue LDLA, magenta = SRCR.

We redrew the figure and improved the quality (450dpi).

(30) "Accordingly, a short parallel β -sheet between the N-terminal segment and the SPD domain was observed in MSPL, whereas the C-terminal end of hepsin forms an antiparallel β -sheet (Fig. 2A)": Unfortunately, I cannot see this interaction in this small panel 2A, may be you can add an inset for this panel which shows the described interaction.

In accordance with this comment, we added a close-up view of the beta-sheet interaction in Fig 2A (red box).

Page 7:

(31) Hence, both MSPL and hepsin may bind substrate in close proximity to the transmembrane region: Replace this by: "Hence, both MSPL and hepsin may bind their substrates in close proximity to the transmembrane region."

In accordance with this comment, we replaced the sentence on page 9, line 7-8.

(32) Third line from bottom: After Arg 556 add the chymotrypsin numbering.

In accordance with this comment, we added "Arg561(Asn245)" on page 9, line 5.

Page 8:

(33) Change the title of the paragraph to: Interaction of the inhibitor decanoyl-RVKR-cmk in the active site of MSPL

In accordance with this comment, we replaced the title (page 9) as "Interaction of the inhibitor decanoyl-RVKR-cmk in the active site of human MSPL" on page 9. For clarity, "MSPL" is replaced by "human MSPL".

(34) replace "S1 family" by "S1A family of serine protease" (see the nomenclature given in the Merops database)

In accordance with this comment, we replaced the "S1 family" by "S1A family" on page 9, line 13.

(35) In the activated MSPL, Ile321 at the cleavage site forms a salt bridge with the conserved Asp505 residue located immediately prior to the catalytic Ser506 residue (Fig. S1A).: change to: "In the activated MSPL, Ile321(16) at the cleavage site forms a salt bridge with the conserved Asp505(194) residue located immediately prior to the catalytic Ser506(195) residue (Fig. S1A)."

In accordance with this comment, we changed the sentence on page 9, line 13-16.

(36) Of course, somewhere at the beginning of the manuscript the authors should provide a hint that always the chymotrypsin numbering is used in round brackets after the full-length number for residues of the SPD.

In accordance with this comment, we added chymotrypsin numbering in brackets after the MSPL residue.

(37) Change "This interaction might be generated by the activating cleavage." To "This interaction is generated by the activating cleavage." Please provide a reference for this common mechanism, which was several times nicely described by Wolfram Bode (e.g., for tPA, although there the zymogen has relatively high activity: Stubbs, M. T.; Rensus, M.; Bode, W. An active zymogen: unravelling the mystery of tissue-type plasminogen activator. Biol Chem 1998, 379, 95-103.) But this paper nicely explains the activation mechanism.

In accordance with this comment, we changed the sentence at page 9, line 16, and added a citation for Stubbs et al, 1998.

(38) "This salt bridge was also observed in other proteases such as plasma kallikrein (20)(PDB entry: 1Z8G) and hepsin.": This you will find for each of the 60/70 trypsin-like serine protease, a little exception is tPA, where the zymogen already possesses considerable activity.

In accordance with this comment, we changed the sentence as follows; "This "Ile16-Asp194 salt-bridge" is a common feature among the trypsin-like proteases (Halfon & Craik, 1998)." and cite the chapter of "Handbook of Proteolytic Enzymes" on page 9, line 19-20.

(39) "A furin inhibitor peptide binds to the SPD of MSPL with P1-Arg, P2-Lys, C-terminal cmk (chloromethylketone; an active site-direct group) and N-terminal decanoylgroup(Fig.1C, 3)." change to "The chloromethyl group of the inhibitor irreversibly alkylates His57 of the SPD of MSPL, in addition, an hemiketal if formed to the active site Ser195. In addition, several interactions via the P1-Arg and P2-Lys side chains are formed (Fig.1C, 3).

In accordance with this comment, we changed the sentence on page 9, line 21-page 10, line 1.

(40) "to the backbone amides of the oxyanion hole (Gly504 and Ser506)" replace by ".....Gly504(193) and Ser506(195)"

In accordance with this comment, we changed the sentence on page 10, line 4-5.

(41) "as a hydrogen bond with the side chain of Ser501 and the backbone carbonyl of Gly5292 should be replace by "as a hydrogen bond with the side chain of Ser501(190) and the backbone carbonyl of Gly529(219)." The authors should check this, but I think these are the numbers of the SPD.

In accordance with this comment, we added chymotrypsin numbering in brackets after the MSPL residue on page 10, line 6-7.

(42) Write: "Asp500(189) is located in the bottom of S1 pocket."

In accordance with this comment, we added the sentence on page 10, line 7-8.

(43) There seem to exist significant errors related to the numbering and text in the following sentence: "...Lys interacts with residues at the so-called 99-loop (99 comes from the chymotrypsinogen numbering!) that contains the catalytic residue Asp409" Therefore, I always suggest to use the SPD numbering, because otherwise the name 99-loop would not make sense if it is located close to residue 409! However, I checked the uniprot entry mentioned before, there residue 409 is a Glu and not Asp! The catalytic Asp102 is number 414 on the full-length numbering in the file I have used. Moreover, I would not say that Asp 102 belongs to the 99 loop, although it is close to residue 99. The whole residue numbers for all comments made in the text and the related sentences have to be revised and corrected, if necessary. May be, I am wrong, but I simply had a look at the uniprot entry. Interestingly, before Asp414(102) there are five (!) additional acidic residues DEEDD (sequence 408-412 corresponding to 96-100 based on chymotrypsinogen).

In accordance with this comment, we revised several points as follows;

- 1) The human MSPL numbering refers to UniProt# Q9BYE2. The previous numbering is shifted by minus five.
- 2) The chymotrypsin numbering is described in parentheses.
- 3) The Asp102(Asp414 in MSPL) is not within the 99-loop. Therefore, we changed the sentence as follows; "However, P2-Lys interacts with basic residues located at the 99-loop (chymotrypsin numbering) next to the catalytic residue Asp414(Asp102)."

Page 9:

(44) "with the backbones of Asp403 and Glu405, the side chains of Tyr401 and Asp406...": also here, the correct chymotrypsin numbering should be added. Again, in the uniprot entry Q9BYE2[1 - 586] I could not find Asp403 and Asp405 with these numbers, at position 401 I have found this sequence:

I401-INSNYTDEE410

In accordance with this comment, we revised several points as follows;

- 1) The human MSPL numbering refers to UniProt Q9BYE2.
- 2) The chymotrypsin numbering is described in parentheses.

(45) chains of Asp406 and the catalytic Asp409 residue: same, again is 409 really the catalytic Asp102, which residue is 406?

We revised the residue numbers to correspond to Q9BYE2. Thus, Asp406 and Asp409 changes to Asp411(Ile99) and Asp414(Asp102), respectively. We added the chymotrypsin numbering in parentheses.

(46) Based on all these trouble with the residue numbers (may be, I am wrong), I suggest to revise Figure 4 and add as first entry the sequence of chymotrypsin (this starts at residue 1), in this case the SPD of all other proteases start at residue 16. By doing this, the authors can easily deduce the appropriate residue numbers.

The chymotrypsin numbering should be also added in brackets for the residues of the SPD in the Ligplot (Figure 3).

In accordance with this comment, we added the chymotrypsin numbering in parentheses.

(47) Two mistakes in the following statement: "... there is no hydrogen bond between the side chains of P3-Val/P4-Lys and the MSPL." Sentence makes no sense and is wrong. It is clear that no H-bond can be formed from a Val side chain! Moreover, the CMK inhibitor possesses an Arg in P4 and not Lys!

In accordance with this comment, we corrected the sentence as follows; "Compared with P1-Arg and P2-Lys, there is no distinct interactions between the side chains of P3-Val/P4-Arg and the human MSPL."

(48) Replace Gly527 by Gly527(216)

We replaced Gly527 with Gly532(Gly216) on page 10, line 21 & 22.

(49) "The side chain of P3-Val makes van der Waals interactions with Trp526 and Gly527". This is very unusual, but I have seen it in the pdb file. Normally, a P3 side chain of a residue in L-configuration goes into the solvent. Moreover, this is a very unusual backbone conformation, in which the P3 NH goes up into the solvent. I all other structures which I know for trypsin-like serine proteases in complex with substrate analog inhibitors this NH goes down to the carbonyl of Gly216, it means, the P3 backbone forms a short antiparallel beta-sheet interaction with Gly216. This very artificial binding mode is probably caused by the presence of the artificial sulfate ion.

In accordance with this comment, we revisited the inhibitor conformation. After reexamination, we conclude that the P3-P4 moiety is likely to be an artifact due to crystal packing (new figure;

Fig 4). As a consequence, we modelled the putative RVKR peptide using the hepsin: ace-KQLR-cmk structure (PDB: 1Z8G) as template. The P4-Arg interaction is discussed in the revised manuscript. We added the putative model in Fig 4. The results are described in the section entitled “Interaction of the inhibitor decanoyl-RVKR-cmk in the active site of MSPL”.

(50) During looking on the pdf file I have also realized an unusual (most-likely wrong) torsion angle for the amide bond between the decanoyl residue and the P4 Arg backbone NH, which looks like approximately 90 degree. I assume, it should be 180 degree (trans amide bond). The authors should check, if there is an error in their structure and revise their pdb file.

As suggested, we reexamined the P4-Arg-dec linkage structure and corrected the mistake. The Ramachandran plot analysis of the corrected structure of P4-Arg is within the favored region (below). The structure will be updated in PDB, and Table S1 has also been updated.

(51) Figure 4, showing the sequence alignment based on the full-length numbering should be also enlarged. Moreover, it would be great to add an additional panel 4B, showing only the alignment of the SPD with chymotrypsin as first line. This would provide the correct numbers for the SPD based on chymotrypsin.

In accordance with this comment, Figure 4 was updated to 600dpi and renumbered to Figure 5. Also, the amino-acid sequence alignment with chymotrypsin is added to the new Figure 4.

(52) Figure 5: Figure is also very tiny and should be enlarged. Moreover, I strongly suggest to show all structures of active site inhibitors for trypsin-like serine proteases or furin-like PCs in the so-called standard orientation, suggested always by W. Bode several decades ago (as example see the 2003 Henrich paper showing the crystal structure of the used inhibitor in complex with furin (PDB: 1P8J), Figure 4, on the right we have the P1 residue, than the peptide backbone goes from

the right to the left side...)

In accordance with this comment, the quality of original Figure 5 (new Figure 6) was improved to 450dpi. We redrew the peptidase: inhibitor figure with standard orientations.

Page 10:

(53) "As a consequence, the P1, P2, and P4 site contacts with furin, whereas the P3 site is directed away from it": A hint should be given that in case of furin the P3 Val backbone makes an important antiparallel beta-sheet interaction with furin residue Gly255 (this is the equivalent residue to Gly216 in proteases of the S1A family), whereas in the MSPL structure the P3 residue makes only one backbone interaction, probably caused by the presence of the artificial sulfate ion.

We found the P3-P4 conformation may be an artifact of crystal packing. Therefore, the P3-P4 interaction of MSPL is discussed with the putative model.

(54) "To date, the structure of TMPRSS2 has not been reported." change to "To date, the experimental structure of TMPRSS2 has not been reported." At least for the SPD, there exist several homology models, but probably not for the full-length extracellular domain.

In accordance with this comment, we changed the sentence on page 13, line 10.

(55) Eight out of nine disulfide bonds are conserved (Fig. 4): It is very hard to see them in this very small Figure 4 showing the alignment. Perhaps, an additional schematic structure for TMPRSS2 could be added (like in Figure 1A, e.g., in the supporting info) in direct comparison to MSPL showing the disulfide bonds (then the disulfide bonds have not to be shown in Figure 1A).

In accordance with this comment, we renewed the original Figure 4 (new Figure 5) in high quality. A schematic representation of TMPRSS2 has been added to Figure 1A.

(56) Furthermore, Glu404, an important residue for P2-Lys recognition in MSPL, is replaced by Lys225 in TMPRSS2 (Fig. 4, 6B): I wonder about these many discrepancies in the residue numbering compared with the uniprot entry for human MSPL. In their crystal structure there is absolute no interaction between Glu404 and the P2 Lys side chain, there exists an interaction between the Glu405 carbonyl oxygen! This error is independent from any confusion regarding the residue numbering. I would appreciate, if the responsible author of this paper would have an own look on the structure!

We apologize for this error. As shown in Figure 3 and the new Figure 5 (sequence alignment), Asp411(Ile99) interacts with N ζ of P2-Lys, and the corresponding residue in TMPRSS2 is Lys225.

(57) I am disappointed about all these mistakes, because it makes a lot of work for a reviewer. Originally, I assume, the authors speak from residue 99 at the tip of the 99 hairpin loop, correct? I

know that this is Lys99 in case of TMPRSS2. Please use the chymotrypsin numbering!

For clarity, we added the chymotrypsin numbering in parentheses.

(58) Figure 6: The Figure has to be improved and enlarged (all panels). The inhibitor should be added to panel A to see, if this is placed close to the region in the red rectangle, where differences between TMPRSS2 and MSPL exist, or if it bound far away (what should be the case).

In accordance with this comment, we renewed original Figure 6 (new Figure 7) in high quality. Moreover, a stick model of tri-peptide (VKR) was added to Figure 7A.

(59) "As mentioned earlier, this substitution leads to a preference for the monobasic target of TMPRSS2. In fact, the S1/S2 cleavage site of SARS-CoV-2 spike protein is reported as P2-Ala instead of a basic residue (25, 26, 27). In summary, our homology model reflects the features of TMPRSS2 target peptide recognition.": This whole paragraph gives a very wrong impression that the S1/2 site in S of SARS-CoV-2 with Ala in P2 position is cleaved by TMPRSS2, which is not the case. This site is cleaved/activated by furin! TMPRSS2 is supposed to cleave at the S2 site of SARS-CoV-2, and this sequence is PSKPSKR|SFIEDL, it means it also has a basic residue in P2 position (see the recent paper, Bestle at al. 2020 in Life Sci Alliance. 2020 Jul 23;3(9):e202000786.).

Sincere apologies - we misunderstood the TMPRSS2 cleavage site. As described in the first paragraph, TMPRSS2 cleaves the S2' site of SARS-CoV-2 S-protein. Thus, we have retracted the conclusion that Lys225 reduces the affinity of TMPRSS2 for the basic residue at P2 site. As shown in Fig 7B, the electrostatic surface potential around the P2-Lys binding site is negatively-charged, suggesting that Lys/Arg at P2 site is acceptable. In addition, we noticed that the characteristic 3D structure of the S-protein S2' site that forms a loop-helix structure where the alpha-helix starts from the P1'-Ser (Figure S2, PDB: 6XR8), requires a wider binding cleft at the P1' position for TMPRSS2. This corresponds to our homology model. These considerations are discussed in the section entitled "Homology model analysis of TMPRSS2".

Page 11, conclusion section:

(60) "HAPI virus" correct to "HPAI virus" or "HPAIV" (it comes from highly pathogenic avian influenza virus.)

We corrected "HAPI virus" to "HPAI virus".

December 7, 2020

Re: Life Science Alliance manuscript #LSA-2020-00849R

Prof. Yuushi Okumura
Sagami Women's University
Department of Nutrition and Health
2-1-1 Bunkyo, Minami
Sagamihara, Kanagawa 252-0383
Japan

Dear Dr. Okumura,

Thank you for submitting your revised manuscript entitled "Crystal structure of inhibitor-bound human MSPL that can activate high pathogenic avian influenza" to Life Science Alliance. The manuscript has been seen by the original reviewers whose comments are appended below. While the reviewers continue to be overall positive about the work in terms of its suitability for Life Science Alliance, some important issues remain.

Our general policy is that papers are considered through only one revision cycle; however, given that the suggested changes are relatively minor, we are open to one additional short round of revision.

As you will see from the reviewers' comments, the numbering between with PDB file and the manuscript is off and needs to be revised. As both reviewers have laid out ways to improve the manuscript for publication in LSA, we would make an exception and would like to invite a second final revision for this manuscript. Please note that full support from the reviewers would be essential to move forward with the revised manuscript. We request you to resubmit a revised final version that addresses all of the reviewers' points.

Please submit the final revision within one month, along with a letter that includes a point by point response to the remaining reviewer comments.

- A letter addressing the reviewers' comments point by point.
- An editable version of the final text (.DOC or .DOCX) is needed for copyediting (no PDFs).
- High-resolution figure, supplementary figure and video files uploaded as individual files: See our

detailed guidelines for preparing your production-ready images, <https://www.life-science-alliance.org/authors>

B. MANUSCRIPT ORGANIZATION AND FORMATTING:

Sincerely,

Shachi Bhatt, Ph.D.
Executive Editor
Life Science Alliance
<https://www.lsjournal.org/>
Tweet @SciBhatt @LSAJournal

Reviewer #1 (Comments to the Authors (Required)):

I read the revised paper. The author adequately addressed my concerns. However, I read reviewer 2's comment and the author's revision. I pay to reviewer 2 comments. Mainly, they pointed the very unusual conformation of p3 and p4. Thus, the author recognized that the original structure is an artificial structure due to the crystal packing. They submitted a new paper based on the model structure of the putative PVKR complex.

The corrected structure is reasonable.
I have claim points as follows.

(1) They resubmitted the new PDB, but the numbering of PDB is not the same as in this paper. S511, H366, and D414 in the new paper. But S506, H361, and D409 in PDB.

Numbering is essential. The author, especially the responsible author, should check directly the coordinate.

(2) Reviewer 2 comment (21)

I appreciated the model PDB file of T MPRSS2. However, their deposited file is PDF. The PDF is not a useful file to look at the PDB coordinate, thus change to the PDB as the supplementary file. Moreover, please mention this in data availability and Fig 7.

(3) Please recheck the whole numbering, including newly deposited PDB.

(4) New Fig4

The author compared the hepsin with ace-KWQLR-cmk (1Z8G).

If so, please use the same direction as Fig.1 B and C.

Show 1Z8G whole fig and close up using the same direction.

Next, show the RVKM putative model using the same direction.

This figure is essential compared with their artificial structure and the RVKM putative model. Thus use understandable (especially for a general biochemist, not protease expert) figures as much as they can.

(5)

Please check the supp2 size. Too large.

(6)

Fig3 ligplot fig is not correct.

Symmetry molecule should be included.

Is so, D482(B) should appear in this figure to interact with P4-Arg.

I am wondering the paper should be acceptable. However, if we consider much work of reviewer 2 as mentioned 56, the paper might be acceptable, but I think the author really should pay significant acknowledgment to reviewer 2.

Reviewer #2 (Comments to the Authors (Required)):

As I had written in my first review, I appreciate very much the interesting and new content of this manuscript. Moreover, the revised manuscript is considerably improved, also the Figures are now more clear. However, several points still have to be improved in a second revision, see below:

Abstract:

Page 2 in pdf, l. 2: remove colon after (HA)

l. 3: better write "convertases" (Plural) and write "type II transmembrane serine proteases (TTSP)" (see Antalis, T. M., Bugge, T. H., and Wu, Q. (2011) Membrane-anchored serine proteases in health and disease, Progress in molecular biology and translational science 99, 1-50).

l. 4: replace "chymotrypsin" by "trypsin", although trypsin-like proteases possess a chymotrypsin-like fold, in contrast to trypsin the protease chymotrypsin cleaves after neutral P1 residues.

l.6: write "or MSPL"

l. 15: I do not understand the final sentence of the abstract that TMPRSS2 can accommodate substrates with an alpha-helix at the P1' position. This must be wrong, because a helix cannot be formed by a single residue. Perhaps the authors want to write P' region. But I have doubt that this is correct, because proteases bind their substrates via extended beta-sheet interactions. The authors should read. (Tyndall, J. D., Nall, T., and Fairlie, D. P. (2005) Proteases universally recognize beta strands in their active sites, Chem Rev 105, 973-999.)

p. 3, l. 9: better write "certain influenza virus HAs"

p. 5, l. 21/22: better write ...with the irreversible peptidic inhibitor decanoyl..... (sounds better)

p. 6, l. 4: better write "...recognize both Lys or Arg as P4 residues" (I would not write 2 x P4)
l. 7-9: As written above, a single position or residue cannot be sufficient for the recognition of an helix, it must be an appropriate region. The authors cannot write S1'. Perhaps they write "S prime region". I simply do not understand their statement.

p. 7, l.4: In Figure 1A the SRCR domain stops at residue 317, here the authors write 318. What is correct? In Figure 1A the SPD start at residue 327, in the text the authors write 326. In the pdb file I have seen Ile326, it seems that the labeling of Figure 1A is slightly wrong or the pdb file is wrongly labeled. Also in the caption to Figure 1 the authors write that the SPD start at residue 327. Moreover, I would appreciate, if the authors can also label the domains of TMPRSS2 and hepsin in Figure 1A with the appropriate residue numbers (it would help the readers to compare these enzymes). Unfortunately, Figure 1C is not in standard orientation. In standard orientation the P1 side chain goes from north east to south west, the catalytic triad is on the right side and Asp189 at the bottom.

l. 7: write "known as furin inhibitor I" (remove a), also "data were collected" (I am not sure)

l. 11: in the pdb file the SPD stops at position 563, therefore I do not understand why the authors write here in text only "range 193-558" on line 11 and on line 6 187-586. This is very confusing.

l. 15: the catalytic part can be abbreviated as SPD.

l.17: here the authors write that the activation cleavage occurs between Arg325-Ile326, therefore it seems that the pdb is correctly labeled, but Figure 1A is wrongly labeled.

l. 20: This is important! I only suggest to provide the chymotrypsin numbering, but not the residues in chymotrypsin (they are not important). Therefore, line 19 should be changed to: ... hereafter, the residue numbers in parentheses denote the corresponding chymotrypsin residue number" (not the name!). Of course, in case of Ile 326 this corresponds to Ile 16, this is ok.

l. 21: write Asp510(194) (delete the second Asp, of course it is also Asp in chymotrypsin or chymotrypsinogen).

l. 23: is there any information if the activation is achieved by an autoactivation or by a different protease? In case of TMPRSS2 and matriptase an autoactivation is always described (in case of an Ser195Ala mutation always the non-cleaved form was found suggesting an autoactivation. I am not sure if this was also tried with MSPL?

p. 9, l. 5: write only Arg561(245) it is not important that residue 245 in chymotrypsin is Asn.

l. 14: Ile326(16)

l. 15: Asp510(194) and Ser511(195) (delete the second Asp and Ser)

l. 19: In Figure 3 I cannot see a hairpin loop, as written in the text. Perhaps the authors find a better place to insert the reference to Figure 3 in the text. In Figure 3A G534 should be labeled as G534(219), also after all other residues the chymotrypsin numbers should be added in brackets in Fig. 3A.

Fig. 3B: in the alignment a free space should be inserted before G534(219) (residue 218 is missing in MSPL) than the following highly conserved Cys220 would be at the correct and identical position in the chymotrypsin and MSPL sequence. There are many proteases like thrombin, factor Xa, plasmin, uPA which all lack residue 218 (some people say that residue 219 is missing, both is ok, but the two cysteines in position 220 should be aligned.

L 22: His366(57)

L 23: Ser511(195)

p. 10, l.5: Gly509(193) and Ser511(195)

l. 6: Asp505(189)

l. 7: Ser506(190)

l 12: Asp414(102)

l 13: Are five H-bonds correct, did you check also all the angles, if they are correct? Asp408(Ser96). I have even seen two water molecules in H-bond distance to the Lys side chain amino group.

I. 14: Glu410(98) and Asp411(99) Please correct this in all the following cases, I will not write this anymore at all positions, it is too much work!

I. 22: I do not understand in which way the P3 Val side chain makes an hydrophobic interaction with Gly532(216). Gly is not a great residue for hydrophobic interactions.

p. 11, l 1: Gln537(221)

I. 9: better write ... to form an antiparallel beta-sheet interaction...

I. 10: Gly532(216)

I. 13. Which residues correspond to Asp427 and Asn539 in chymotrypsin?

L 19: Glu409(97) and Glu410(98)

L 20: Asp411(99) and Tyr489(175)

p. 12, l. 15: The P3 side chain is directed into the solvent (the P3 backbone of Val in furin makes the same antiparallel beta-sheet interaction with Gly255 in furin, like Gly216 in trypsin-like proteases)

I. 22-24: delete the sentences with the K_i values, they are completely wrong. Irreversible inhibitors should not be characterized by K_i values, they should be characterized by the term k_{inact}/K_i , which corresponds to a second order inactivation rate constants. Of course, you can also determine a K_i for irreversible inhibitors based on the hyperbolic dependency of the k_{obs} values from the inhibitor concentration, but this was not done in these papers (simply have a look at: Stein, R. L., and Trainor, D. A. (1986) Mechanism of inactivation of human leukocyte elastase by a chloromethyl ketone: kinetic and solvent isotope effect studies, *Biochemistry* 25, 5414-5419 or look in a text book for enzyme kinetics, e.g. from Copeland. This is very helpful). But it is not ok to deduce a K_i from IC_{50} measurements, because the strength of inhibition simply depends from the length of the reaction of the irreversible inhibitor with the enzyme. If you have a higher inhibitor concentration than the enzyme concentration and you wait long enough, the enzyme becomes always dead.

Reply to Reviewer #1

First of all, we would like to thank the reviewer as we found his/her comments and suggestions constructive in improving our manuscript. Following are our responses to these comments and suggestions.

(1) They resubmitted the new PDB, but the numbering of PDB is not the same as in this paper. S511, H366, and D414 in the new paper. But S506, H361, and D409 in PDB.

Numbering is essential. The author, especially the responsible author, should check directly the coordinate.

- We are sorry for delay in the release of the revised PDB. The revised coordinate of 6kD5 have been released at 25th Nov 2020.

(2) Reviewer 2 comment (21)

I appreciated the model PDB file of Tmprss2. However, their deposited file is PDF.

The PDF is not a useful file to look at the PDB coordinate, thus change to the PDB as the supplementary file. Moreover, please mention this in data availability and Fig 7.

- The text file was converted to PDF automatically, due to the specifications of the submission system. We asked the editor not to convert it. We added the sentence of "The coordinate of the homology model of human Tmprss2 is available from supplementary materials." in Figure 7 caption.

(3) Please recheck the whole numbering, including newly deposited PDB.

-We are sorry for the mistakes. The discrepancy of MSPL numbering found in the figures and the captions, were fixed.

(4) New Fig4

The author compared the hepsin with ace-KWQLR-cmk (1Z8G).

If so, please use the same direction as Fig.1 B and C.

Show 1Z8G whole fig and close up using the same direction.

Next, show the RVKM putative model using the same direction.

This figure is essential compared with their artificial structure and the RVKM putative model. Thus use understandable (especially for a general biochemist, not protease expert) figures as much as they can.

- We replaced the Fig1C with a standard orientation diagram as the Reviewer #2's comment. We added a supplemental figure (new Fig S2) including; (A) Hepsin/acetyl-KQLR-cmk (same direction as Fig1B), (B) A close-up view of acetyl-KQLR-cmk (same direction as Fig1C), (C) Superposition of acetyl-KQLR-cmk and decanoyl-KVCR-cmk (same direction as Fig4B and Fig6D), and (D) Superposition of acetyl-KQLR-cmk and putative model of KVCR-cmk (same direction as Fig4C).

(5)

Please check the supp2 size. Too large.

- According to the reviewer #2's comment, we deleted the speculation that TMPRSS2 can cleave alpha-helical structure at P1'-P11' region. Along with it, we withdrawn the figure S2 and replaced by a new figure S2 that shows acetyl-KQLR-cmk(1Z8G) and RVCR putative model.

(6)

Fig3 ligplot fig is not correct.

Symmetry molecule should be included.

Is so, D482(B) should appear in this figure to interact with P4-Arg.

- Following the comment, we added Asp482 to Fig 3A.

I am wondering the paper should be acceptable. However, if we consider much work of reviewer 2 as mentioned 56, the paper might be acceptable, but I think the author really should pay significant acknowledgment to reviewer 2.

- We are grateful to the reviewer #2 for the cordial comments, and described our gratitude in this response letter.

Reply to Reviewer #2

First of all, we would like to sincerely thank the reviewer #2 for the cordial comments. Following are our responses to these comments and suggestions.

Abstract:

(1) Page 2 in pdf, l. 2: remove colon after (HA)

- Following the comment, we removed the colon.

(2) l. 3: better write "convertases" (Plural) and write "type II transmembrane serine proteases (TTSP)"

(see Antalis, T. M., Bugge, T. H., and Wu, Q. (2011) Membrane-anchored serine proteases in health and disease, Progress in molecular biology and translational science 99, 1-50).

- Following the comment, we rewrote the sentence as "... such as proprotein convertases and type II transmembrane serine proteases (TTSP)."

(3) l. 4: replace "chymotrypsin" by "trypsin", although trypsin-like proteases possess a chymotrypsin-like fold, in contrast to trypsin the protease chymotrypsin cleaves after neutral P1 residues.

- Following the comment, we rewrote the sentence as "... is cleaved by trypsin-like proteases, ..." in abstract.

(4) l.6: write "or MSPL"

- Following the comment, we replaced "... and MSPL" by "... or MSPL" in abstract.

(5) l. 15: I do not understand the final sentence of the abstract that TMPRSS2 can accommodate substrates with an alpha-helix at the P1' position. This must be wrong, because a helix cannot be formed by a single residue. Perhaps the authors want to write P' region. But I have doubt that this is correct, because proteases bind their substrates via extended beta-sheet interactions. The authors should read. (Tyndall, J. D., Nall, T., and Fairlie, D. P. (2005) Proteases universally recognize beta strands in their active sites, Chem Rev 105, 973-999.)

- We had speculated that TMPRSS2 could cleave the target peptide having an α -helix at P1'-P11' in the previous manuscript, but we have no evidence at all, so we removed this sentence.

(6) p. 3, l. 9: better write "certain influenza virus HAs"

- Following the comment, we replaced "influenza virus hemagglutinin" by "certain influenza virus HAs" in introduction.

(7) p. 5, l. 21/22: better write ...with the irreversible peptidic inhibitor decanoyl..... (sounds better)

- Following the comment, we replaced "... human MSPL in complex with the irreversible inhibitor decanoyl-RVKR-cmk peptide at 2.6 Å resolution." by "... human MSPL in complex with the irreversible peptidic inhibitor decanoyl-RVKR-cmk at 2.6 Å resolution." in introduction.

(8) p. 6, l. 4: better write ...recognize both Lys or Arg as P4 residues" (I would not write 2 x P4)

- Following the comment, we replaced "... recognize both the P4-Lys and P4-Arg." by "... recognize both Lys or Arg as P4 residues." in introduction.

(9) l. 7-9: As written above, a single position or residue cannot be sufficient for the recognition of an helix, it must be an appropriate region. The authors cannot write S1'. Perhaps they write "S prime region". I simply do not understand their statement.

- As we described above, we removed the speculation that TMPRSS2 could cleave the target peptide forming an α -helix at P1'-P11'. And we rewrote as "The human TMPRSS2 model reveals a wide binding cleft at the S1' position, suggesting that TMPRSS2 can capture the target peptides of flexible conformations."

(10) p. 7, l.4: In Figure 1A the SRCR domain stops at residue 317, here the authors write 318. What is correct? In Figure 1A the SPD start at residue 327, in the text the authors write 326. In the pdb file I have seen Ile326, it seems that the labeling of Figure 1A is slightly wrong or the pdb file is wrongly labeled. Also in the caption to Figure 1 the authors write that the SPD start at residue 327. Moreover, I would appreciate, if the authors can also label the domains of TMPRSS2 and hepsin in Figure 1A with the appropriate residue numbers (it would help the readers to compare these enzymes). Unfortunately, Figure 1C is not in standard orientation. In standard orientation the P1

side chain goes from north east to south west, the catalytic triad is on the right side and Asp189 at the bottom.

- We are sorry for the discrepancy of MSPL numbering. The MSPL subdomains are; LDLA (203-226) SRCR (227-317), and SPD (326-561). We corrected the fig1, the caption of fig1, and the main text. Furthermore, following the comment, we added the domain labeling to fig1A Tmprss2 and Hepsin, and we replaced fig1C by 3D stereo-viewed with standard orientation.

(11) l. 7: write "known as furin inhibitor I" (remove a), also "data were collected" (I am not sure)

- Following the comment, we corrected the sentences.

(12) l. 11: in the pdb file the SPD stops at position 563, therefore I do not understand why the authors write here in text only "range 193-558" on line 11 and on line 6 187-586. This is very confusing.

- Following the comment, we corrected the sentences as "193-563". We also added "Residues of 187-192, 324-325, and 564-586 regions were missing due to disorder." In page 7, line 11-12.

(13) l. 15: the catalytic part can be abbreviated as SPD.

- Following the comment, we corrected as SPD

(14) l.17: here the authors write that the activation cleavage occurs between Arg325-Ile326, therefore it seems that the pdb is correctly labeled, but Figure 1A is wrongly labeled.

- We are sorry for the discrepancy of MSPL numbering. We corrected the fig1, the caption of fig1, and the main text.

(21) l. 20: This is important! I only suggest to provide the chymotrypsin numbering, but not the residues in chymotrypsin (they are not important). Therefore, line 19 should be changed to: ... hereafter, the residue numbers in parentheses denote the corresponding chymotrypsin residue number" (not the name!). Of course, in case of Ile 326 this corresponds to Ile 16, this is ok.

- Following the comment, we described as chymotrypsin number only in the plackets.

(22) l. 21: write Asp510(194) (delete the second Asp, of course it is also Asp in chymotrypsin or chymotrypsinogen).

- Following the comment, we described as chymotrypsin number only in the plackets.

(23) l. 23: is there any information if the activation is achieved by an autoactivation or by a different protease? In case of TMPRSS2 and matriptase an autoactivation is always described (in case of an Ser195Ala mutation always the non-cleaved form was found suggesting an autoactivation. I am not sure if this was also tried with MSPL?

- It is quite interesting topic but we have no experimental data to show whether MSPL has autoactivated or not.

(24) p. 9, l. 5: write only Arg561(245) it is not important that residue 245 in chymotrypsin is Asn.

(25) l. 14: Ile326(16)

(26) l. 15: Asp510(194) and Ser511(195) (delete the second Asp and Ser)

- Following the comments, we described as chymotrypsin number only in the plackets.

(27) l. 19: In Figure 3 I cannot see a hairpin loop, as written in the text. Perhaps the authors find a better place to insert the reference to Figure 3 in the text. In Figure 3A G534 should be labeled as G534(219), also after all other residues the chymotrypsin numbers should be added in brackets in Fig. 3A.

- We are sorry for the mistakes in writing. As the reviewer #2 have pointed out, we have to cite the paper describing the activation mechanisms of zymogen. Here, we cite the review article by Khan & James. We rewrite here as "... hairpin loop (189-loop) (Khan & James, 1998)."

Following the comment, we added chymotrypsin number in all residues in fig3A.

(28) Fig. 3B: in the alignment a free space should be inserted before G534(219) (residue 218 is missing in MSPL) than the following highly conserved Cys220 would be at the correct and identical position in the chymotrypsin and MSPL sequence. There are many proteases like thrombin, factor

Xa, plasmin, uPA which all lack residue 218 (some people say that residue 219 is missing, both is ok, but the two cysteines in position 220 should be aligned.

- We agree that disulfide-bonded Cys should be matched in the sequence alignment, we redrawn fig3B.

(29) L 22: His366(57)

(30) L 23: Ser511(195)

(31) p. 10, l.5: Gly509(193) and Ser511(195)

(32) l. 6: Asp505(189)

(33) l. 7: Ser506(190)

(34) l 12: Asp414(102)

- Following the comments, we described as chymotrypsin number only in the plackets.

(35) l 13: Are five H-bonds correct, did you check also all the angles, if they are correct? Asp408(Ser96). I have even seen two water molecules in H-bind distance to the Lys side chain amino group.

- We have looked carefully the interaction between N ζ in P2-Lys and MSPL, each distance is measured as water732 (3.1 Å), OH of Y406 (2.8 Å), O of D408 (3.0 Å), O of E410 (2.7 Å) and O of D411 (3.0 Å). Among them, Y406 and water732 are the hydrogen donors to the P2-Lys, and D408, E410 and D411 are the acceptors from P2-Lys. The distance between water736 and N ζ is 3.9 Å, and the angle of C ϵ -N ζ -O is 56.7°, therefore it seems unlikely to form a hydrogen bonding. Thus, we consider that there are five hydrogen bonds.

(36) l. 14: Glu410(98) and Asp411(99) Please correct this in all the following cases, I will not write this anymore at all positions, it is too much work!

- Following the comment, we described as chymotrypsin number only in the plackets.

(37) l. 22: I do not understand in which way the P3 Val side chain makes a hydrophobic interaction with Gly532(216). Gly is not a great residue for hydrophobic interactions.

- We agree that Gly C α atom has no hydrophobicity. We remove the words “and Gly532(216)” in line20. In accordance with this change, we corrected the ligplot drawing (fig3A).

(38) p. 11, l 1: Gln537(221)

- Following the comment, we described as chymotrypsin number only in the plackets.

(39) l. 9: better write ... to form an antiparallel beta-sheet interaction...

- Following the comment, we repladed “... to form a β -sheet interaction...” by “... to form an antiparallel β -sheet interaction...”

(40) l. 10: Gly532(216)

- Following the comment, we described as chymotrypsin number only in the plackets.

(41) l. 13. Which residues correspond to Asp427 and Asn539 in chymotrypsin?

- Following the comment, we added chymotrypsin numberings as “Asp472(160)”. Furthermore, “and Asn539” is deleted because the distance between Asn539 and P4-Arg changed longer than the effective interaction.

(42) L 19: Glu409(97) and Glu410(98)

(43) L 20: Asp411(99) and Tyr489(175)

- Following the comments, we described as chymotrypsin number only in the plackets.

(44) p. 12, l. 15: The P3 side chain is directed into the solvent (the P3 backbone of Val in furin makes the same antiparallel beta-sheet interaction with Gly255 in furin, like Gly216 in trypsin-like proteases)

- Following the comment, we replaced “... whereas the P3 site is directed away from it.” by “... whereas the P3 side chain is directed into the solvent.”.

(45) I. 22-24: delete the sentences with the K_i values, they are completely wrong. Irreversible inhibitors should not be characterized by K_i values, they should be characterized by the term k_{inact}/K_i , which corresponds to a second order inactivation rate constants. Of course, you can also determine a K_i for irreversible inhibitors based on the hyperbolic dependency of the k_{obs} values from the inhibitor concentration, but this was not done in these papers (simply have a look at: Stein, R. L., and Trainor, D. A. (1986) Mechanism of inactivation of human leukocyte elastase by a chloromethyl ketone: kinetic and solvent isotope effect studies, *Biochemistry* 25, 5414-5419 or look in a text book for enzyme kinetics, e.g. from Copeland. This is very helpful). But it is not ok to deduce a K_i from IC_{50} measurements, because the strength of inhibition simply depends from the length of the reaction of the irreversible inhibitor with the enzyme. If you have a higher inhibitor concentration than the enzyme concentration and you wait long enough, the enzyme becomes always dead.

- We agree that the irreversible inhibitors should be characterized by k_{inact}/K_i , instead of K_i . However, the K_i value of cmk-peptides are widely used in many papers, and the values we mentioned in the manuscript is taken from the previous papers, so that we cannot recalculate k_{inact}/K_i . We refer to the K_i value to compare the difference in affinity between MSPL and Furin, and we believe that the use of the K_i value is meaningful to this extent.

March 9, 2021

RE: Life Science Alliance Manuscript #LSA-2020-00849RR

Prof. Yuushi Okumura
Sagami Women's University
Department of Nutrition and Health
2-1-1 Bunkyo, Minami
Sagamihara, Kanagawa 252-0383
Japan

Dear Dr. Okumura,

Thank you for submitting your revised manuscript entitled "Crystal structure of inhibitor-bound human MSPL that can activate high pathogenic avian influenza". We would be happy to publish your paper in Life Science Alliance pending final revisions necessary to meet the minor request of Reviewer 2 and our formatting guidelines.

Along with the points listed below, please also attend to the following,

- please consult our manuscript preparation guidelines <https://www.life-science-alliance.org/manuscript-prep> and make sure your manuscript sections are in the correct order; - please separate the Results and Discussion section into two - 1. Results 2. Discussion, as per our formatting requirements
- please make sure the author order in your manuscript and our system match and that there is no name discrepancy in the presentation of the names of your co-authors (e.g. Shigetada Teshima-Kondo in MS file vs. Shigatada Teshima-Kondo in the system)
- please add ORCID ID for the corresponding author-you should have received instructions on how to do so
- please add callouts for Figures 3B; 6A-C; S2A-D to your main manuscript text
- please add your main, supplementary figures, and table legends to the main manuscript text after the reference section, and please upload your Table in editable .doc or excel format as a single file accordingly
- please use the [10 author names, et al.] format in your references (i.e. limit the author names to the first 10)

A. FINAL FILES:

B. MANUSCRIPT ORGANIZATION AND FORMATTING:

Sincerely,

Shachi Bhatt, Ph.D.

Executive Editor

Life Science Alliance

<https://www.lsajournal.org/>

Interested in an editorial career? EMBO Solutions is hiring a Scientific Editor to join the international Life Science Alliance team. Find out more here -

https://www.embo.org/documents/jobs/Vacancy_Notice_Scientific_editor_LSA.pdf

Reviewer #1 (Comments to the Authors (Required)):

All of my concerns were addressed properly.

Reviewer #2 (Comments to the Authors (Required)):

I appreciate the effort of the authors to prepare this significantly improved/corrected manuscript. I still have only one minor hint/comment.

The authors should simply delete the final two wrong sentences on page 12 in their submitted pdf-file, describing K_i values of this irreversible inhibitor for furin and MSPL and the following conclusion (lines 22-24: The K_i value of decanoyl-RVKR-cmk for MSPL (Okumura et al, 2010) and furin (Jean et al, 1998) is 2.9 nM and 0.6 nM, respectively. These K_i values might reflect the differences in P4-Arg affinity.)

These two wrong sentences are not necessary for this otherwise very nice paper and therefore, can be deleted. They want to publish a structure paper, not a paper describing the potency of this inhibitor, what was wrongly done by other authors before.

In their response letter the authors had written: "We agree that the irreversible inhibitors should be characterized by k_{inact}/K_i , instead of K_i . However, the K_i value of cmk-peptides are widely used in many papers, and the values we mentioned in the manuscript is taken from the previous papers, so that we cannot recalculate k_{inact}/K_i . We refer to the K_i value to compare the difference in affinity between MSPL and Furin, and we believe that the use of the K_i value is meaningful to this extent.

Why writing wrong stuff, if the authors know or agree to my comment that this is wrong for an irreversible inhibitor?

Please see the following sentences from the Enzyme kinetic book from Ross Stein (Kinetics of enzyme action, Wiley 2011, Chapter 6.4, page 138)

...Since enzyme and the final complex formed between enzyme and irreversible inhibitor can never attain equilibrium, the only methods that are appropriate to estimate the potency of an irreversible inhibitor are kinetic methods. IC_{50} values determined for irreversible inhibitors have no meaning and will decrease in magnitude as preincubation time between enzyme and inhibitor is lengthened.

Kinetic parameters for enzyme inactivation by an irreversible inhibitor are determined by one of two methods:

- A discontinuous method can be used in which residual enzyme activity is measured after enzyme and inhibitor have incubated for a preset time. These velocities can then be plotted

versus time and fit to Equation 6.41 to arrive at a value of k_{obs} for that particular inhibitor concentration. Or,

- Progress curves can be generated by monitoring product formation after addition of enzyme to a solution of substrate and inhibitor. The progress curve can then be fit to Equation 6.42 to estimate k_{obs} .

In either of the above two cases, k_{obs} is determined at various concentrations of inhibitor to determine the mechanism of inactivation and then the kinetic parameters.

In the provided reference Okumura from 2010 the authors had written:

To determine the effects of inhibitors, enzyme preparations were preincubated for 5 min with various inhibitors at 37°C and the residual enzyme activity was measured.

Why not waiting 10 min or 20 min or 2 h? For each longer time you will get a stronger inhibition.

March 12, 2021

RE: Life Science Alliance Manuscript #LSA-2020-00849RRR

Prof. Yuushi Okumura
Sagami Women's University
Department of Nutrition and Health
2-1-1 Bunkyo, Minami-ku
Sagamihara, Kanagawa 252-0383
Japan

Dear Dr. Okumura,

Thank you for submitting your Research Article entitled "Crystal structure of inhibitor-bound human MSPL that can activate high pathogenic avian influenza". It is a pleasure to let you know that your manuscript is now accepted for publication in Life Science Alliance. Congratulations on this interesting work.

DISTRIBUTION OF MATERIALS:

Again, congratulations on a very nice paper. I hope you found the review process to be constructive and are pleased with how the manuscript was handled editorially. We look forward to future exciting submissions from your lab.

Sincerely,

Shachi Bhatt, Ph.D.

Executive Editor

Life Science Alliance

<https://www.lsjournal.org/>

Interested in an editorial career? EMBO Solutions is hiring a Scientific Editor to join the international Life Science Alliance team. Find out more here -

https://www.embo.org/documents/jobs/Vacancy_Notice_Scientific_editor_LSA.pdf